# CryptoX : Compositional Reasoning Evaluation Framework of Large Language Models

## Abstract

The compositional reasoning ability has long been regarded as critical to the generalization and intelligence emergence (20; 42) of large language models (**LLMs**). However, despite numerous reasoning-related benchmarks (12; 3; 24), the compositional reasoning capacity of LLMs is rarely studied or quantified in the existing benchmarks. In this paper, we introduce **CryptoX**, a plug-in evaluation framework that to quantify the compositional reasoning capacity of LLMs. Building upon CryptoX, we construct **CryptoBench**, which integrates atomic transformation rules from CryptoX into a set of relatively simple benchmarks, which serve as proxies to reflect models' CR ability in tackling more complex real-world problems. We conduct comprehensive experiments on 40+ widely used LLMs using CryptoBench with the well-designed metric, which clearly show disparities in their CR abilities. Through further analytical experiments, we demonstrate that CryptoX can indeed evaluate the true CR ability of models. Moreover, by analyzing open-source models with mechanistic interpretability methods, we find that the CR process exhibits a clear stage-wise structure—Subtask Decomposition, Subtask Solving, and Integration. Finally, through both formal analysis and experiments, we show that two of these stages, corresponding to Reasoning Path Planning Ability and Subtask Decomposition Ability, play a pivotal role in determining the effectiveness of the CR process.

## 1 Introduction

Compositional reasoning (**CR**) refers to the ability to break down complex problems into simpler components and then use those components to form new ideas *within a single model run*—conceptualized as combining ability A and ability B simultaneously (denoted $A + B$, see Figure 1(a)) (18). Existing research works (42; 18; 4) have pointed out that compositional reasoning plays a key role in the generalization and intelligence emergence of LLMs. Quantifying the compositional reasoning ability and behavior of LLMs helps reveal how they transfer knowledge and skills from pretraining and alignment data to solve new problems, and uncover patterns of emergent generalization (15) as the size of LLMs increases.

However, existing reasoning-related benchmarks are either tightly coupled to specific domains (6; 16; 13) or pursue orthogonality of pretraining knowledge (12). Zhao et al. (48), for instance, introduces compositional relation reasoning, yet their tasks are limited to simple, narrow-domain relational reasoning, failing to reflect broader CR abilities. KORBench (24) primarily assesses reasoning with knowledge orthogonal to pre-training rather than directly addressing compositional reasoning. Furthermore, the multi-hop reasoning explored by Yang et al. (46), which is based on entity substitution in relational triples $(e, r, e)$, uses overly simplistic and narrowly focused tasks, which prove insufficient for capturing genuine CR. As a result, despite numerous existing reasoning-related benchmarks, the CR abilities of LLMs have not been well studied or quantified (18).

However, the CR process in real-world scenarios is often complex, diverse, and difficult to quantify. Inspired by cryptographic techniques (26), we select a set of transformations and encryption methods and abstract them into **instruction encryption** and **instruction transformation**, which serve as proxies to reflect models' CR ability in tackling more complex real-world problems, leading to the development of **CryptoX**[1].

---

[1] https://anonymous.4open.science/r/CryptoX-CF44

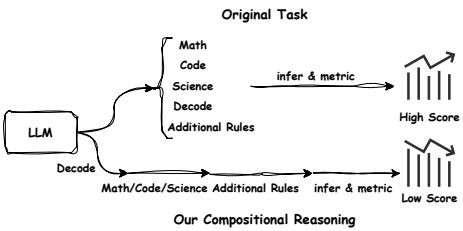
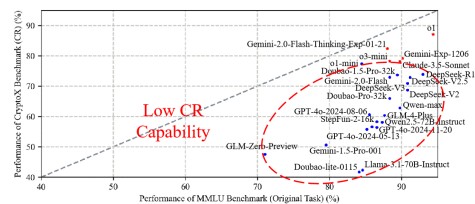

(a) Concept of compositional reasoning

(b) Performance gap between original benchmarks and CryptoBench

Figure 1: (a) shows the concept of compositional reasoning (CR) which involves combining different abilities in one single model run (e.g., A+B) instead of reasoning via individual ability (e.g., A **or** B). (b) shows the evaluation result of CryptoX. Some LLMs with strong reasoning abilities on the original benchmark have the low CR abilities on CryptoBench.

**Instruction encryption** randomly encodes part of each instruction in the benchmarks using a given codebook. **Instruction transformation** defines additional projection rules for modifying segments of words within each instruction. For instance, applying the Repeat Transformation to the term "happy" yields the modified form "hhaappppyy".

Subsequently, we combine CryptoX with representative benchmarks from different domains, specifically those where models achieve relatively high base performance. These benchmarks are then integrated into a systematic evaluation framework to assess models' CR ability. Based on detailed experiments with 40+ LLMs, we propose AUC as an efficient evaluation metric for reliably assessing models' true CR ability. The results reveal that most existing LLMs exhibit weak CR performance, and consistently yield a similar trend of performance decay across different domains. We then conduct three complementary analyses and obtain the following findings: We then conduct three complementary analyses: (1) Through experiments on transformation complexity, multi-turn vs. single-turn comparisons, and cross-benchmark correlations, we demonstrate that CryptoX faithfully measures models' compositional reasoning ability while avoiding the high cost and overfitting issues of similar benchmarks. (2)During the process of solving CR tasks, models exhibit a clear stage-wise reasoning pattern, which can be broadly divided into three stages: Subtask Decomposition, Subtask Solving, and Integration. (3)Reasoning Path Planning Ability and Subtask Decomposition Ability emerge as the primary factors determining a model's CR capability. We further formalize these dimensions as parameters $\alpha$ and $\beta$, enabling quantitative analysis of CR ability across models.

The contributions of this paper are as follows:

- We propose CryptoX, a flexible evaluation method for LLMs' compositional reasoning.

- We build CryptoBench and evaluate LLMs' compositional reasoning ability based on it.

- We demonstrate that the CR process can be divided into three stages—Subtask Decomposition, Subtask Solving, and Integration—and that CR ability can be further characterized along two dimensions: Reasoning Path Planning Ability and Subtask Decomposition Ability, which we quantify using the parameters $\alpha$ and $\beta$.

## 2 CRYPTOX: COMPOSITIONAL REASONING EVALUATION METHOD

### 2.1 OVERVIEW

CryptoX presents a novel approach to evaluate LLMs' CR ability by integrating existing benchmarks with cipher-based transformations. The method first encodes specific words in prompts into new ciphered characters and explicitly includes encoding rules within instructions. To further increase difficulty, some instructions require multi-step reasoning, guiding models to follow structured solution steps. LLMs must first decode ciphered characters before answering the original questions, ensuring a rigorous assessment of CR. Building upon this framework, we construct CryptoBench, a benchmark set that systematically applies these principles for evaluation.

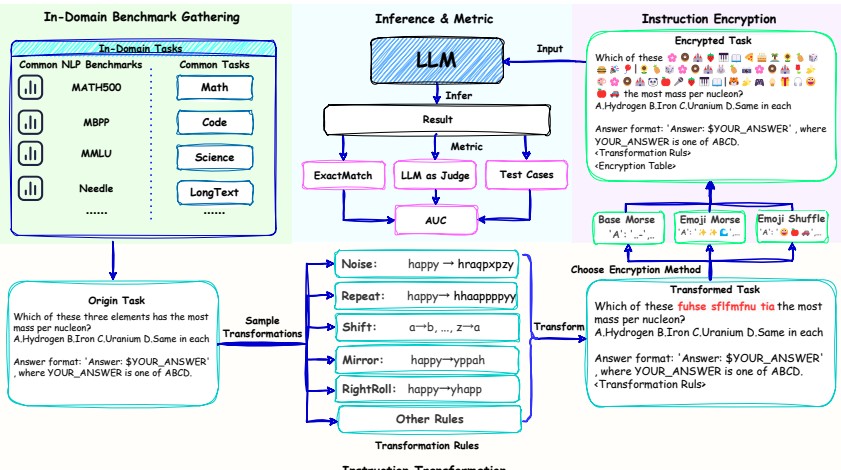

Figure 2: Overview of the CryptoBench Construction Process. We apply instruction encryption and transformation to the tasks from common NLP benchmarks and combine them to construct our CryptoBench Task. Then we use Exact Match, LLM as judge, UnitTest and AUC as our Evaluation Metrics to judge LLMs' performance.

## 2.2 TASK DEFINITION AND METHODOLOGY

Given a benchmark $B = \{x_1, x_2, \ldots, x_N\}$, where $x_i$ is a prompt consisting of an instruction and a question $Q$, and $B$ contains $N$ data points. To transform the existing prompt into a compositional reasoning prompt, we define a set of transformation rules, where each data point undergoes a series of transformations. Specifically, the transformation process can be described as:

$$x' = r_m \circ r_{m-1} \circ \cdots \circ r_1(x), \tag{1}$$

where $r_i$ denotes the $i^{\text{th}}$ transformation rule, and $\circ$ represents the composition of transformations. Based on our formal definitions, we explore two implementation approaches of $r$ : **instruction encryption**, which encodes parts of the prompt, and **instruction transformation**, which restructures it. These transformations enable the creation of diverse compositional reasoning tasks.

---

**Algorithm 1** Instruction Encryption

**Input:** Original question $Q = (w_1, \ldots, w_n)$, where $w_i$ are words split by spaces, encoding mapping table $\mathcal{M}$, and the number of encoded words $m$
**Output:** Encoded question $Q_e$
$selected\_words \leftarrow []$
**for** 1 to $m$ **do**
    $w_r \leftarrow \text{RandomSelect}(Q)$
    **if** $|w_r| > 1 \wedge w_r \notin selected\_words$ **then**
        $w_r \leftarrow \mathcal{M}(w_r)$
        $selected\_words.\text{append}(w_r)$
    **end if**
**end for**

---

**Instruction Encryption** Specifically, as illustrated in Figure 2, Instruction Encryption applies three encoding rules to the input prompt; its detailed process is presented in Algorithm 1. This mechanism facilitates the encoding of prompts containing any number of words, ensuring that the resulting encoded representations, generated according to user-defined rules, do not overlap with the pre-training corpus. For simplicity, we adopt the emoji-shuffle encoding rule as an example in the following sections; a comprehensive description of all encoding rules is provided in Appendix D.1.

**Instruction Transformation** The vanilla approach involve prompting the LLM to provide an answer in the format of "Answer: A" for the multi-choice question tasks. To further evaluate the compositional reasoning abilities of LLMs in handling OOD scenarios, as shown in Figure 2, we establish instruction transformation to further increase the number of reasoning hops. (1) **Noisy**

**Rule**: Noise is added to words by inserting a random character after every character located at an odd position (1st, 3rd, 5th, etc.). For example, "happy" becomes "hiapjpyk". (2) **Difficult Transformation Rule**: This rule incorporates seven distinct transformations. For instance, one such transformation, termed "Repeat", involves duplicating each letter within a word (e.g., "happy" will be "hhaappppyy"). Another, "Shift", alters each letter by systematically changing it to a different one based on a positional adjustment (e.g. "a" might become "b", "b" could become "c", etc.).

## 3 EXPERIMENT

Table 1: **Statistics of our Benchmarks**: Avg. Len refers to the total number of characters in each question with 0-, 5-, or 10-word encoding numbers, respectively. Answer Format includes the following types: ME (Mathematical Expression), SC (Single Choice), CB (Code Blocks), TE (Textual Expression), and MC (Multiple Choices).

| Category | Total Nums | Avg. Len | Ans. Fmt |
|---|---|---|---|
| Crypto-Math | 500 | 441.89 / 1410.86 / 1471.17 | ME |
| Crypto-MMLU | 285 | 627.97 / 1274.82 / 1333.6 | SC |
| Crypto-MMLU-Num | 285 | 699.97 / 1346.82 / 1405.6 | SC |
| Crypto-MMLU-Alpha | 285 | 922.97 / 1569.82 / 1628.6 | SC |
| Crypto-MBPP | 427 | 621.17 / 1268.54 / 1327.14 | CB |
| Crypto-BBH | 405 | 1585.3 / 3464.98 / 3517.24 | TE&MC |
| Crypto-Needle-30K | 100 | 64811.41 / 63535.62 / 62111.8 | TE |
| Crypto-HighResolution | 497 | 1177.49 / 2360.61 / 2426.62 | ME&SC&TE&MC&CB |

(1) **LLMs**. We evaluate both open-source and closed-source models on CryptoBench. For closed-source LLMs, we evaluate GPT series (29) (GPT-4o), Claude series (1), and o1 series (30), and the like. For open-source LLMs, we evaluate Qwen2.5 series (40), Llama-3.1 series (8), Codestral (39) and Jamba-1.5-mini (19), and the like.

(2) **Vanilla Benchmark**. As shown in Table 1, we apply our method to five benchmarks: MATH (16), MMLU (14), BBH (38), MBPP (2), and Needle (11). For MATH, we use the MATH 500 subset; for MMLU, we adopt MMLU-dev; for MBPP, we adopt MBPP-sanitized; and for Needle, we use the three-needle setting (Only needle is encoded). Benchmarks are evaluated at three Instruction Encryption (IE) levels: 0, 5, and 10, where IE=0 represents the unmodified baseline. "Crypto-HighResolution" is created by sampling these benchmarks, applying IE values from 0 to 10.

(3) **Prompt Setting**. To systematically evaluate the models under various experimental settings, we design the following prompt templates (c.f., Appendix D.1): a. **Zero-shot Prompts** are applied on subsets including Crypto-Needle-30K, Crypto-Math, Crypto-MBPP, and Crypto-MMLU. b. **Few-shot Prompts** are applied to the Crypto-BBH subset.

(4) **Evaluation Metrics**. a. **Exact Match**: For simple answers, we use regular expressions to extract and normalize the model's answer, then compare to the the correct answer. b. **LLM as judge**: For complex outputs (e.g., math),Judger LLM evaluates against reference solutions.(Detailed setting in Appendix D.4) c. **UnitTest**: For code, we run provided tests and score by pass rate: The number of passed test cases is then counted, and the final score is calculated as: score = (passed test cases) / (total test cases).

(5) **AUC of Compositional Reasoning**. To better assess a model's overall performance across different difficulty levels, we compute an area under the curve (AUC) score. After obtaining the evaluation metrics, we plot the number of encoded words $k$ as the x-axis and the corresponding model performance as the y-axis. The AUC is then calculated using the trapezoidal rule:

$$\text{AUC} = \int_{k_{\min}}^{k_{\max}} f(k)\, dk \approx \sum_{i=1}^{N-1} (k_{i+1} - k_i) \frac{y_i + y_{i+1}}{2} \tag{2}$$

where $k_i$ represents the number of encoded words (Instruction Encryption level), and $y_i$ is the corresponding model performance. A higher AUC indicates enhanced compositional reasoning ability with diverse instruction encryption levels.

### 3.1 MAIN RESULTS

Table 2 presents the performance of diverse models on our CryptoBench across various domains.

Table 2: Performance of different models on CryptoBench. 0 / 5 / 10 represents the number of words encoded. Values with the highest, second-highest, and third-highest accuracies are marked in green , blue , and orange respectively.

| Model | AUC | Avg | Crypto-Math | | | Crypto-MBPP | | | Crypto-BBH | | | Crypto-MMLU | | | Crypto-MMLU-Num | | | Crypto-MMLU-Alpha | | | Crypto-Needle-30K | | |
|---|---|---|---|---|---|---|---|---|---|---|---|---|---|---|---|---|---|---|---|---|---|---|---|
| | | | 0 | 5 | 10 | 0 | 5 | 10 | 0 | 5 | 10 | 0 | 5 | 10 | 0 | 5 | 10 | 0 | 5 | 10 | 0 | 5 | 10 |
| *Closed-source LLMs* | | | | | | | | | | | | | | | | | | | | | | | |
| o1 | 4.05 | 83.69 | 96.99 | 89.66 | 84.48 | 64.93 | 68.04 | 63.9 | 84.08 | 83.66 | 82.13 | 94.35 | 92.25 | 90.53 | 92.5 | 91.43 | 90.0 | 90.07 | 87.99 | 87.32 | 99.66 | 80.33 | 43.2 |
| Claude-3.7-Sonnet | 3.95 | 81.41 | 78.6 | 73.92 | 68.24 | 69.16 | 69.82 | 72.14 | 86.67 | 83.78 | 80.52 | 91.93 | 90.55 | 85.55 | 90.88 | 90.88 | 84.41 | 87.72 | 86.1 | 83.27 | 98.67 | 84.22 | 52.67 |
| Gemini-2.5-Pro-Preview-0506 | 3.93 | 78.46 | 94.6 | 87.8 | 90.2 | 68.7 | 79.39 | 77.44 | 71.73 | 72.22 | 72.59 | 89.47 | 92.98 | 92.98 | 62.22 | 61.52 | 61.29 | 61.52 | 61.29 | 60.82 | 100.0 | 98.67 | 90.33 |
| Claude-3.7-Sonnet-Thinking | 3.93 | 79.03 | 90.0 | 84.6 | 85.8 | 88.63 | 89.62 | 90.09 | 71.6 | 71.6 | 72.47 | 90.29 | 91.58 | 90.99 | 59.18 | 60.59 | 60.35 | 61.05 | 59.41 | 60.12 | 99.67 | 95.67 | 86.33 |
| o3-mini-high | 3.75 | 77.58 | 98.2 | 88.2 | 85.84 | 59.09 | 63.66 | 61.86 | 81.38 | 67.43 | 63.08 | 90.18 | 88.92 | 87.83 | 89.12 | 88.6 | 86.69 | 87.72 | 84.36 | 83.27 | 74.0 | 58.33 | 41.33 |
| o3-mini | 3.67 | 76.38 | 95.99 | 85.4 | 77.62 | 46.96 | 60.39 | 57.49 | 80.6 | 72.03 | 72.7 | 88.34 | 86.97 | 84.91 | 88.93 | 88.57 | 86.07 | 87.23 | 85.87 | 83.1 | 88.55 | 52.0 | 34.35 |
| Gemini-2.0-Flash-Thinking | 3.58 | 76.06 | 87.98 | 78.3 | 73.79 | 63.31 | 51.05 | 47.05 | 82.34 | 84.65 | 82.63 | 87.99 | 88.03 | 83.86 | 91.07 | 89.29 | 85.36 | 84.75 | 83.04 | 81.34 | 98.32 | 51.0 | 22.11 |
| Gemini-2.5-Flash-Preview-0417 | 3.58 | 73.24 | 90.59 | 84.85 | 78.12 | 68.8 | 77.3 | 75.44 | 64.99 | 62.5 | 60.74 | 89.68 | 92.31 | 89.92 | 61.15 | 60.15 | 59.95 | 60.38 | 59.34 | 58.52 | 99.29 | 81.0 | 62.92 |
| GPT4.1-0414 | 3.54 | 73.14 | 83.0 | 73.6 | 67.6 | 78.91 | 84.76 | 85.0 | 71.48 | 70.49 | 70.0 | 91.23 | 89.83 | 89.83 | 60.82 | 59.18 | 58.95 | 58.48 | 57.54 | 56.61 | 100.0 | 84.67 | 44.0 |
| Gemini-Exp-1206 | 3.52 | 74.72 | 85.57 | 73.22 | 68.14 | 53.3 | 56.15 | 53.99 | 79.85 | 80.45 | 76.18 | 90.11 | 84.51 | 77.9 | 92.14 | 83.93 | 80.0 | 85.82 | 79.15 | 73.94 | 97.64 | 60.67 | 36.39 |
| Claude-3.5-Sonnet | 3.45 | 74.07 | 74.75 | 66.33 | 60.69 | 69.18 | 66.93 | 67.54 | 84.58 | 81.68 | 77.42 | 89.75 | 86.97 | 83.51 | 90.36 | 82.5 | 81.07 | 82.98 | 77.74 | 78.17 | 98.99 | 38.0 | 16.33 |
| o1-mini | 3.43 | 73.98 | 90.78 | 78.3 | 66.33 | 72.69 | 65.37 | 67.88 | 83.33 | 77.23 | 75.93 | 84.45 | 80.99 | 79.3 | 85.36 | 77.86 | 80.36 | 80.5 | 73.5 | 74.3 | 95.96 | 35.0 | 28.23 |
| o3-mini-low | 3.39 | 73.31 | 96.0 | 77.53 | 66.74 | 57.23 | 65.65 | 62.24 | 81.38 | 69.73 | 66.86 | 87.37 | 84.69 | 81.37 | 86.32 | 85.02 | 80.61 | 82.1 | 80.46 | 67.3 | 99.67 | 40.0 | 21.33 |
| o4-mini-0416 | 3.28 | 68.71 | 98.4 | 90.8 | 86.2 | 57.92 | 59.25 | 59.33 | 68.77 | 56.67 | 58.39 | 92.75 | 91.23 | 91.23 | 60.82 | 60.59 | 60.59 | 61.05 | 58.95 | 59.88 | 83.33 | 52.33 | 34.33 |
| Gemini-2.0-Flash | 3.14 | 68.62 | 89.18 | 65.11 | 58.87 | 49.65 | 50.86 | 46.37 | 79.1 | 72.28 | 68.73 | 88.34 | 79.93 | 77.54 | 87.14 | 79.29 | 72.14 | 73.05 | 74.56 | 66.55 | 96.63 | 44.0 | 21.77 |
| Doubao-1.5-Pro-32k | 2.94 | 66.22 | 81.76 | 61.46 | 49.4 | 72.99 | 64.13 | 59.44 | 85.25 | 71.71 | 89.4 | 80.63 | 75.79 | 90.36 | 78.21 | 68.57 | 53.19 | 56.54 | 50.7 | 95.62 | 28.0 | 3.4 | |
| GPT-4o-2024-08-06 | 2.73 | 61.9 | 75.95 | 47.67 | 36.49 | 48.03 | 52.1 | 47.47 | 81.09 | 70.3 | 64.52 | 85.51 | 78.17 | 75.09 | 83.93 | 71.07 | 65.0 | 63.12 | 51.24 | 37.68 | 97.98 | 42.0 | 25.51 |
| Doubao-Pro-32k | 2.64 | 62.32 | 87.58 | 52.94 | 36.49 | 64.64 | 65.7 | 60.05 | 81.84 | 74.26 | 62.78 | 88.34 | 72.18 | 64.21 | 87.14 | 75.0 | 64.29 | 60.64 | 53.71 | 47.89 | 97.98 | 11.0 | 0.0 |
| GPT-4o-2024-05-13 | 2.6 | 60.4 | 73.95 | 44.42 | 29.23 | 49.33 | 58.53 | 50.33 | 80.6 | 69.06 | 57.82 | 85.16 | 76.06 | 70.17 | 84.29 | 64.64 | 54.64 | 68.79 | 52.3 | 36.97 | 97.31 | 38.33 | 26.53 |
| Qwen-max | 2.57 | 61.48 | 81.16 | 45.03 | 27.82 | 59.48 | 55.27 | 46.67 | 81.84 | 71.54 | 58.56 | 89.75 | 73.94 | 64.91 | 89.29 | 73.57 | 64.64 | 81.92 | 62.19 | 52.46 | 88.55 | 16.0 | 6.46 |
| GPT-4o-2024-11-20 | 2.55 | 57.91 | 68.94 | 47.87 | 35.08 | 41.27 | 25.88 | 37.14 | 80.74 | 70.55 | 63.52 | 86.57 | 77.82 | 71.23 | 73.57 | 68.93 | 63.93 | 51.77 | 44.88 | 38.38 | 99.33 | 43.33 | 24.83 |
| GLM-4-Plus | 2.53 | 59.29 | 71.34 | 44.83 | 29.64 | 45.09 | 57.79 | 51.07 | 81.34 | 69.55 | 58.56 | 87.63 | 72.54 | 60.0 | 80.0 | 71.43 | 62.14 | 74.47 | 66.08 | 52.82 | 97.64 | 8.0 | 3.06 |
| StepFun-2-16k | 2.35 | 52.81 | 75.75 | 48.48 | 32.46 | 58.55 | 57.88 | 52.8 | 88.81 | 82.18 | 65.51 | 86.57 | 70.42 | 60.7 | 83.93 | 63.57 | 57.5 | 37.59 | 31.45 | 25.7 | 20.2 | 8.33 | 0.68 |
| Gemini-1.5-Pro-001 | 2.09 | 49.06 | 64.53 | 39.96 | 27.02 | 42.07 | 50.83 | 43.84 | 60.95 | 51.98 | 47.39 | 79.51 | 60.91 | 58.6 | 73.93 | 69.29 | 57.86 | 58.51 | 41.34 | 31.34 | 66.67 | 3.67 | 0.0 |
| GPT-4-Turbo | 2.07 | 52.48 | 70.34 | 38.34 | 21.57 | 43.76 | 36.53 | 21.13 | 81.09 | 66.83 | 55.34 | 85.87 | 69.01 | 56.14 | 80.71 | 57.86 | 47.14 | 62.41 | 50.18 | 45.07 | 98.65 | 9.33 | 4.76 |
| GLM-Zero-Preview | 2.06 | 48.89 | 69.34 | 31.03 | 18.55 | 45.65 | 39.84 | 37.7 | 73.88 | 62.87 | 48.39 | 71.03 | 63.38 | 54.74 | 74.29 | 66.79 | 50.71 | 57.8 | 55.48 | 43.66 | 57.91 | 2.33 | 1.36 |
| Doubao-lite-0115 | 1.55 | 41.34 | 74.35 | 32.66 | 16.13 | 43.4 | 39.37 | 29.57 | 72.64 | 56.68 | 46.15 | 84.1 | 63.73 | 52.98 | 71.07 | 42.14 | 37.5 | 4.25 | 3.53 | 4.58 | 84.85 | 5.67 | 2.72 |
| *Open-source LLMs* | | | | | | | | | | | | | | | | | | | | | | | |
| DeepSeek-R1-Distill-Llama-70B | 3.29 | 68.97 | 88.8 | 67.0 | 49.0 | 67.04 | 68.1 | 65.37 | 84.44 | 79.75 | 73.33 | 90.88 | 87.72 | 80.7 | 87.02 | 83.16 | 74.39 | 34.03 | 29.47 | 31.23 | / | / | / |
| DeepSeek-R1 | 3.2 | 68.1 | 90.38 | 78.91 | 70.36 | 68.53 | 70.61 | 69.35 | 86.0 | 80.45 | 78.66 | 92.93 | 90.14 | 90.88 | 61.43 | 42.86 | 37.5 | 46.81 | 45.94 | 45.77 | 87.88 | 59.67 | 35.03 |
| DeepSeek-V2 | 3.08 | 68.29 | 85.37 | 64.5 | 54.23 | 60.83 | 57.86 | 57.28 | 84.08 | 77.23 | 67.49 | 90.81 | 80.99 | 75.79 | 90.36 | 80.71 | 73.93 | 71.99 | 64.31 | 58.8 | 87.54 | 31.33 | 18.71 |
| DeepSeek-V2.5 | 3.08 | 68.23 | 85.37 | 65.31 | 56.65 | 61.45 | 61.66 | 57.93 | 82.84 | 74.5 | 68.73 | 91.17 | 79.22 | 74.74 | 88.93 | 77.86 | 75.0 | 75.89 | 63.96 | 57.75 | 86.19 | 28.67 | 19.05 |
| DeepSeek-V3-1226 | 3.07 | 68.35 | 85.97 | 66.94 | 55.44 | 61.95 | 57.34 | 58.01 | 83.08 | 77.72 | 67.0 | 90.81 | 80.99 | 77.9 | 90.36 | 79.29 | 72.86 | 75.89 | 60.42 | 57.75 | 87.54 | 27.0 | 21.09 |
| DeepSeek-R1-Distill-Qwen-32B | 2.95 | 66.24 | 91.0 | 55.4 | 26.4 | 67.55 | 66.49 | 66.84 | 83.95 | 73.09 | 58.52 | 91.23 | 77.19 | 63.51 | 88.77 | 70.53 | 61.05 | 60.7 | 50.18 | 40.0 | / | / | / |
| DeepSeek-R1-Distill-Qwen-14B | 2.53 | 58.51 | 90.0 | 44.2 | 20.0 | 69.02 | 68.36 | 65.53 | 80.74 | 65.43 | 54.82 | 87.02 | 67.37 | 59.65 | 82.46 | 65.96 | 51.93 | 36.84 | 23.86 | 20.0 | / | / | / |
| Qwen2.5-72B-Instruct | 2.44 | 59.99 | 82.6 | 41.0 | 23.8 | 66.2 | 54.58 | 43.35 | 81.48 | 68.15 | 55.56 | 88.07 | 73.68 | 60.7 | 90.18 | 75.09 | 62.81 | 70.88 | 56.14 | 47.37 | 97.78 | 12.59 | 7.78 |
| Qwen2.5-72B | 2.18 | 48.83 | 52.0 | 22.6 | 9.6 | 66.51 | 59.04 | 52.91 | 55.56 | 42.96 | 35.31 | 59.65 | 57.89 | 43.86 | 65.61 | 58.25 | 44.91 | 56.49 | 54.03 | 41.75 | / | / | / |
| DeepSeek-R1-Distill-Llama-8B | 1.82 | 43.27 | 84.2 | 30.4 | 13.4 | 59.09 | 59.24 | 55.11 | 69.63 | 55.31 | 47.16 | 76.14 | 54.74 | 41.05 | 45.97 | 38.95 | 29.12 | 6.67 | 6.32 | 6.32 | / | / | / |
| Llama-3.1-70B-Instruct | 1.74 | 44.65 | 56.2 | 27.6 | 15.2 | 50.27 | 48.59 | 44.84 | 68.39 | 59.01 | 52.59 | 84.56 | 70.53 | 55.44 | 54.03 | 36.49 | 31.23 | 43.51 | 23.51 | 17.54 | 94.44 | 3.7 | 0.0 |
| DeepSeek-R1-Distill-Qwen-7B | 1.29 | 35.54 | 89.6 | 20.2 | 8.4 | 60.21 | 53.27 | 51.3 | 71.11 | 40.25 | 27.16 | 69.83 | 39.65 | 34.39 | 30.88 | 16.84 | 11.93 | 9.47 | 3.51 | 1.75 | / | / | / |
| Qwen2.5-7B-Instruct | 1.24 | 34.56 | 48.2 | 14.6 | 4.8 | 50.06 | 14.29 | 7.21 | 60.99 | 44.2 | 33.09 | 78.25 | 51.58 | 40.7 | 70.17 | 40.7 | 34.03 | 17.89 | 7.37 | 3.86 | / | / | / |
| DeepSeek-R1-Distill-Qwen-1.5B | 0.89 | 24.22 | 70.2 | 20.2 | 5.6 | 39.7 | 30.8 | 27.87 | 47.41 | 27.9 | 19.26 | 47.37 | 26.32 | 23.16 | 15.79 | 13.33 | 17.89 | 1.75 | 0.7 | 0.7 | / | / | / |
| Codestral-22B-V0.1 | 0.83 | 22.09 | 32.2 | 9.4 | 2.2 | 60.34 | 37.47 | 22.13 | 40.99 | 26.91 | 22.47 | 32.63 | 26.67 | 21.4 | 19.3 | 6.79 | 7.72 | 12.3 | 4.56 | 3.16 | / | / | / |
| Qwen2.5-7B | 0.8 | 20.03 | 22.8 | 8.4 | 3.8 | 44.07 | 30.29 | 27.84 | 25.43 | 17.28 | 13.33 | 36.49 | 20.7 | 22.46 | 29.47 | 18.95 | 18.6 | 9.47 | 8.07 | 3.16 | / | / | / |
| Llama-3.1-8B-Instruct | 0.73 | 22.86 | 34.0 | 7.0 | 3.0 | 33.83 | 11.78 | 8.88 | 37.78 | 20.74 | 18.02 | 65.96 | 31.93 | 26.67 | 55.09 | 25.26 | 19.3 | 9.83 | 1.05 | 1.4 | / | / | / |
| AI21-Jamba-1.5-mini | 0.57 | 16.28 | 29.4 | 3.2 | 2.6 | 52.7 | 18.64 | 5.97 | 28.89 | 21.98 | 22.22 | 39.65 | 24.91 | 29.12 | 6.7 | 3.86 | 3.16 | 0.0 | 0.0 | 0.0 | / | / | / |
| Qwen2.5-1.5B-Instruct | 0.39 | 14.45 | 31.6 | 4.4 | 1.4 | 41.88 | 5.7 | 1.62 | 26.17 | 20.74 | 17.78 | 52.63 | 17.19 | 16.14 | 13.3 | 4.91 | 3.51 | 0.7 | 0.0 | 0.35 | / | / | / |
| Qwen2.5-1.5B | 0.07 | 2.46 | 4.4 | 0.2 | 0.2 | 7.44 | 0.55 | 0.47 | 4.69 | 5.43 | 2.72 | 7.37 | 2.1 | 1.05 | 3.86 | 1.75 | 1.75 | 0.35 | 0.0 | 0.0 | / | / | / |

(1) **Current Models Demonstrate Limited CR Abilities**. As shown in Table 2, increasing the number of encoded words consistently raises reasoning difficulty and lowers evaluation metrics for all models. The fact that this pattern holds across diverse domains and benchmarks suggests a systematic issue rather than a domain-specific one, thereby demonstrating that CryptoX reliably reflects models' true CR ability even on the easiest benchmarks. Beyond this general trend, we further observe that although most models perform poorly on compositional reasoning tasks (see Appendix K), o1, Claude-3.7-Sonnet, and Gemini-2.5-Pro-Preview-0506 still achieve relatively strong performance. Open-source models, however, remain generally weaker than closed-source ones, with a substantial accuracy gap in question answering.

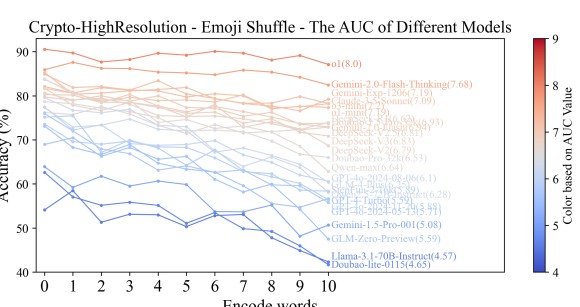

Figure 3: The performance of different models on *Crypto-HighResolution*.

(2) **CR Ability can be Better Evaluated Using AUC**. Conventional benchmarks (e.g., MMLU) now show saturated, narrow accuracy gaps that obscure true ability. Therefore, we adopt AUC as a more discriminative measure of CR. In Figure 3, the redder the lines, the higher the corresponding AUC

values, which also indicate stronger CR abilities of the model. It can be observed that o1, Claude-3.7, and Gemini-2.5-Pro exhibit relatively strong CR abilities. Furthermore, variance of the models' AUC when evaluated in our CryptoX settings is significantly larger than vanilla settings, as illustrated in Appendix E.8,and thus highlights the robustness and sensitivity of AUC.

Table 3: Model's accuracy is compared across different transformation rules applied to a Math-based dataset, with 10 encoded words.

| Model | Avg | Encoding Rule | Noisy Rule + Encoding Rule | Difficult Transformation Rule + Encoding Rule |
|---|---|---|---|---|
| Gemini-2.5-Pro-Preview-0506 | 55.2 | 90.2 | 47.0 | 28.4 |
| DeepSeek-R1 | 43.72 | 70.36 | 34.8 | 26.0 |
| Claude-3.7-Sonnet | 42.34 | 68.24 | 38.6 | 20.2 |

## 4 ANALYSIS

### 4.1 RQ1:DOES CRYPTOX TRULY CAPTURE THE COMPOSITIONAL REASONING (CR) ABILITY OF MODELS?

We conducted additional experiments to further verify the validity of our simple approach in measuring a model's CR capacity.

(1) **Consistent Evaluation Result Under Easy and Hard Transformations**. To assess CryptoX's generality, we test it with other rules (Noisy, Difficult Transformation; see Appendix D). As shown in Table 3, increasing the difficulty of these transformations reduces the absolute accuracy of models on combinatorial reasoning tasks (e.g., R1 drops from 70% to 26%). However, this change has little impact on the relative ranking of models, which indicates that CryptoX captures models' true CR ability simply and directly, without being confounded by the complexity of transformation rules.

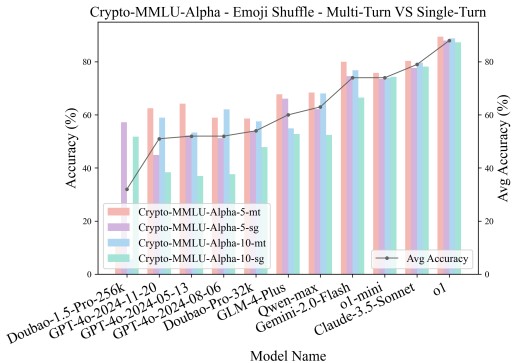

Figure 4: The performance of models with different model sizes. In *Domain_Words*, *Words* denotes the number of words encoded in given question.

(2) **CryptoX Captures $A + B$ Rather Than $A \times B$**. To verify that CryptoX measures models' ability on *single-turn compositional reasoning* $(A + B)$, rather than on the mere product of two independent tasks $(A \times B)$, we conduct the following experiment. (i) **Multi-Turn**: first decode, then answer; (ii) **Single-Turn**: decode and answer simultaneously under the encoding rules. As shown in Figure 4, most models satisfy $\mathrm{acc}(A) \times \mathrm{acc}(B) > \mathrm{acc}(A + B)$, confirming that CryptoX captures true single-turn compositional reasoning ability rather than simple task multiplication.

(3) **Our Benchmark's Distinctiveness from Similar Benchmarks**. To examine how CryptoX differs from existing evaluations, we conduct a correlation study with 18 models across KORBench, ZebraLogic, and other benchmarks. As shown in Figure 5, CryptoBench exhibits strong but not perfect correlation with logic reasoning benchmarks (0.81 0.85 Spearman correlation), indicating that it measures related yet not identical abilities with transferable CR skills beyond string-level operations to semantic-level tasks. Moreover, unlike these specialized benchmarks, CryptoBench is a plug-in method with substantially lower evaluation cost.

### 4.2 RQ2: HOW CAN WE INTERPRET COMPOSITIONAL REASONING AS A PROCESS IN CRYPTOX?

We further conduct interpretability analyses on Crypto-MMLU using two complementary methods: Logit Lens and Neuron Activation Analysis to uncover underlying mechanisms of LLMs' CR pability.

(1) **Logit Lens**. Logit lens (27) is a technique used to interpret the intermediate outputs of language models by mapping hidden states directly to the output vocabulary. In logit lens analysis experiments, we define two target sets, $T_1$ and $T_2$.

$T_1$ consists of the options and content of the answers, while $T_2$ consists of the decoded content from the encoded elements in the question. $T_1 = \bigcup_{a \in \mathcal{A}} \sigma(a)$ and $T_2 = \bigcup_{w \in \mathcal{E}} \sigma(\mathcal{D}(w))$, where $\mathcal{D}(w)$ denotes the correct decoding process using our codetable. $\mathcal{A}$ denotes the answer set and $\mathcal{E}$ denotes the encoded words set. $\sigma(.)$ denotes the function that transforms the original word into a series of candidate prefix tokens with different tokenization. For example, $\sigma(``water'') = [``wa'', \ldots, ``water'']$. When the question and encoding rules are provided as input to the LLM, we extract the hidden state of the last token $h_e$ during the reasoning process and then get the word distribution $p_e$ through the word projection. Then, we accumulate the probabilities of all candidate tokens from $T_1$ or $T_2$ to get the total probability:
$p = \sum_{v \in T_1 \text{ or } v \in T_2} \text{Softmax}(h_i W_p)[v]$, where $p$ is denoted as logits lens.

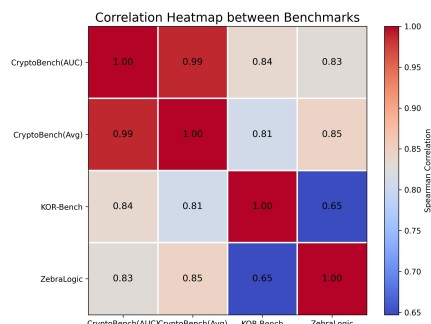

Figure 5: The correlation matrix for benchmarks. The Spearman correlation is ranging from -1 to 1, where 1 indicates stronger alignment.

(2) **Neuron Activation Analysis.** Neuron Activation Analysis examines the activation patterns of individual neurons within LLMs to understand their contributions during reasoning. Specifically, our experiment selects Base Morse as the encoding rule and categorizes the tokens into two sets: (1)**Vocab** contains tokens about the encoding rules (e.g. 'A': '.−', 'B':

Table 4: Subtask Decomposition: scores and $\alpha$ on Crypto-MMLU. $P_1 = P(Q \to A \mid K, I), P_2 = P(Q \to A \mid K)$

| Model | $P_1$ | $P_2$ | $\alpha$ |
| --- | --- | --- | --- |
| Claude-3.5-Sonnet | 67.52 | 63.86 (−3.66%) | 0.945 |
| GPT-4o-1120 | 41.56 | 32.98 (−8.58%) | 0.793 |
| DeepSeek-R1 | 77.17 | 77.03 (−0.14%) | 0.998 |
| Doubao1.5-pro-thinking | 78.95 | 77.19 (−1.76%) | 0.977 |

'−...',...). (2)**Encoded** contains encoded tokens of the question (e.g. $.. - .|.. - | - .| - . - . . . .$). We extract the activation values of the neurons in the MLP layers of LLM during reasoning and then normalize them to a range of 0 to 10. For each neuron, we examine the activation values of the tokens in the Vocab and Encoded set. If the activation value of a token in this range exceeds 7, we classify that neuron as highly activated and record the number of them in each layer.

The interpretability experiments are illustrated in Figure 6. The top two subfigures present the results of Logit Lens analysis, while the bottom two subfigures display the results of the Neuron Activation Analysis. Based on these results, LLMs' CR process can be broadly divided into three stages:

- **Stage 1: Subtask Decomposition.** The model first observes and analyzes useful information in the problem and decomposes the task into several subproblems. In our experiments, this corresponds to the initial phase of the Neuron Activation Analysis, where attention to both the Vocab and Encoded sets increases.

- **Stage 2: Subtask Solving.** The model then solves the subproblems sequentially according to the planned reasoning path. This is evidenced by the two distinct peaks in the Logit Lens analysis and by the significant rise in activations for the Vocab set in the Neuron Activation Analysis. These results indicate that the model first leverages the codebook information to decode the ciphertext and subsequently answers the decoded question.

- **Stage 3: Integration for Final Conclusion.** Finally, the model integrates the solutions of all subproblems to produce the final answer to the CR task. In our experiments, this stage corresponds to the increase in logits over the decoded content set and the concurrent decrease in neuron activations for the Vocab set.

### 4.3 RQ3: Are Decomposition and Solving the Core Abilities in CR?

Based on the 3 stages identified in RQ2, we find that Subtask Decomposition and Subtask Solving play a crucial role in determining a model's CR ability. Accordingly, we further refine CR ability into

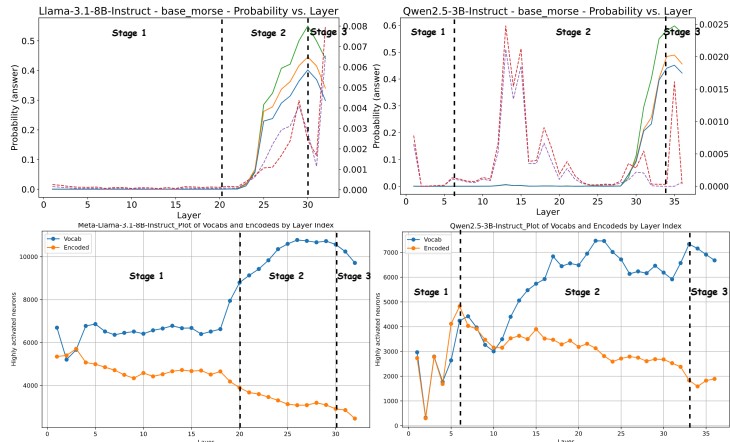

Figure 6: The top two subfigures present the results of the Logit Lens experiments, while the bottom two subfigures show the results of the Neuron Activation Analysis. The green valid line "——" corresponds to the answer set when encoding 0 words; The orange valid line "——" corresponds to the answer set when encoding 3 words; The blue valid line "——" corresponds to the answer set when encoding 5 words; The purple dashed line "- -" corresponds to the decoded words set when encoding 3 words; The red dashed line "- -" corresponds to the decoded words set when encoding 5 words;

two key components: **Reasoning Path Planning Ability** and **Subtask Decomposition Ability**, and then conduct the following ablation experiments:

**Setting**  For a task within CRYPTOBENCH, the required reasoning information comprises:

$Q$: A compositional reasoning problem generated through CryptoX extensions.

$A$: The corresponding answer to the compositional reasoning problem $Q$.

$K$: The complete set of knowledge elements within CryptoX, including all transformation rules, encryption schemes, and noise generation rules.

$K_{\text{sub}}$: A minimal subset of knowledge, $K \supset K_{\text{sub}}$, required to solve $Q$.

$I$: Effective instructions that guide the LLM to solve the problem.

$\rightarrow$: The reasoning process through which the LLM progressively derives the answer $A$ from the question $Q$.

$P$: The final score of the answer set $A$ obtained by LLM.

**(1)Subtask Decomposition Experiment**

We quantify the model's ability to decompose subtasks using the *Task Decomposition Ability Coefficient* $\alpha$:

$$\alpha = 1 - \frac{P(Q \rightarrow A \mid K_{\text{sub}}, I) - P(Q \rightarrow A \mid K)}{P(Q \rightarrow A \mid K_{\text{sub}}, I)} = \frac{P(Q \rightarrow A \mid K)}{P(Q \rightarrow A \mid K_{\text{sub}}, I)}, \quad \alpha \in [0, 1]. \quad (3)$$

Based on this definition, we perform an ablation study using Crypto-MMLU in Table 4.

**(2)Reasoning Path Planning Ability**

**Directed Graph.**  Let the encrypted question be denoted as $Q$ and the decrypted question as $P$. We define a directed graph $G$ whose nodes include the start node $Q$, the end node $P$, and intermediate nodes $T_1, T_2, \ldots, T_n$. Each directed edge corresponds to a decoding rule function $f_i$.

**Path Set.**  Let $\mathcal{P} = \{p^{(1)}, p^{(2)}, \ldots, p^{(K)}\}$ be the set of simple paths from $Q$ to $P$. Each path $p^{(k)}$ traverses a sequence of intermediate nodes:

$$p^{(k)}: \quad Q \xrightarrow{f_{k,1}} T_1 \xrightarrow{f_{k,2}} T_2 \cdots \xrightarrow{f_{k,n_k}} P. \quad (4)$$

**Multi-path Decryption.** For each path $p^{(k)}$, define the composite mapping

$$F^{(k)} = f_{k,n_k} \circ f_{k,n_k-1} \circ \cdots \circ f_{k,1}, \tag{5}$$

and along that path, decryption is given by

$$P = F^{(k)}(Q), \qquad k = 1, 2, \ldots, K. \tag{6}$$

**Optimal Path and Metric $\beta$.** Among all $p \in \mathcal{P}$, there exists an optimal path $p_{\text{opt}}$ containing fewer nodes than all other paths. We include the directed graph as part of the background knowledge $K$ and redefine $K$ and $K_{\text{opt}}$ as follows:

- $K$: The directed graph containing all the paths described above.
- $K_{\text{opt}}$: The subgraph containing only the optimal path $p_{\text{opt}}$.

We define LLM's ability to select the optimal reasoning path using $\beta$:

$$\beta = \frac{P(Q \to A \mid K)}{P(Q \to A \mid K_{\text{opt}})}, \qquad \beta \in [0, 1]. \tag{7}$$

**Experimental Settings.** To evaluate $P(Q \to A \mid K_{\text{opt}})$, we use MMLU as the base dataset and apply encoding at 50% and 100% ratios with a randomly selected transformation rule. The decoding process contains only one optimal path with two mapping steps. To evaluate $P(Q \to A \mid K)$, we again use MMLU with 50% and 100% encodings and a randomly selected transformation rule. The decoding process includes multiple paths with overlapping nodes; the optimal path involves two mapping steps and each remaining path involves three mapping steps.

Table 5: Reasoning Path Planning: scores and $\beta$ on Crypto-MMLU. $P'_1 = P(Q \to A \mid K_{\text{opt}})$, $P'_2 = P(Q \to A \mid K)$.

| Model | $\mathbf{P'_1}$ | $\mathbf{P'_2}$ | $\beta$ |
|---|---|---|---|
| Claude-3.5-Sonnet | 75.44 | 70.53 (–6.5%) | 0.934 |
| Doubao1.5-pro-thinking | 80.35 | 51.93 (–35.3%) | 0.646 |
| DeepSeek-R1 | 77.19 | 73.68 (–3.51%) | 0.954 |
| GPT-4o-1120 | 33.68 | 24.21 (–9.47%) | 0.718 |

**Observations.** The results indicate that:

- Models with higher values of $\alpha$ and $\beta$ tend to achieve higher scores on CryptoX, indicating that Subtask Decomposition Ability and Reasoning Path Planning Ability jointly play a decisive role in determining CR performance.
- The $\alpha$ and $\beta$ of Reasoning-oriented models generally outperform non-reasoning models, suggesting stronger CR ability.

## 5 RELATED WORK

To systematically evaluate the reasoning abilities of LLMs (47; 8; 43; 45; 44; 23), researchers have developed benchmarks encompassing multiple cognitive dimensions, including commonsense reasoning (32; 37; 28), multi-hop reasoning (17; 36; 41), and logical reasoning (35; 13; 25). However, LLMs still face severe challenges when faced with CR tasks that require cross-domain integration (9; 33). Early researches in CR primarily employ CoT with compositional queries to guide model reasoning (49; 7). Existing studies on CR largely utilize multi-hop reasoning benchmarks to investigate models' internal reasoning mechanisms (34; 10; 21).

## 6 CONCLUSION

In this paper, we propose a bench-free compositional reasoning task generation framework CryptoX and develop a CR evaluation benchmark CryptoBench. Based on detailed experiments with 40+ LLMs, we propose AUC as an efficient evaluation metric for reliably assessing models' true CR ability. Through further analytical experiments, we demonstrate that CryptoX can indeed evaluate the true CR ability of models.In addition, we identify three stages in the CR process, among which the first two play a more critical role. From these, we abstract two key sub-abilities that have the greatest impact on CR and conduct a quantitative analysis of them.

## 7 ETHICS STATEMENT

We adhere to the ICLR Code of Ethics. All experiments use publicly available or properly licensed datasets and do not involve intervention on human subjects or the collection of personally identifiable or sensitive information. We comply with data and model licenses, maintain privacy and security for any auxiliary resources, and disclose that no undisclosed conflicts of interest or sponsor influence affected the study's design, execution, or conclusions.

## 8 REPRODUCIBILITY STATEMENT

We provide all experimental settings in the main text and appendix, together with anonymized source code, configuration files, and evaluation scripts, along with detailed usage instructions. This ensures that reviewers and readers can independently reproduce the results by following the referenced sections and supplementary materials.

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

## A  STATEMENT ON LLM USAGE

In this work, LLMs are used only for partial language polishing of the manuscript. All core experiments and conclusions are conducted and derived without the use of LLMs.

## B  LIMITATIONS

CryptoBench effectively evaluates textual compositional reasoning, it currently does not extend to multimodal tasks involving images, audio, or video. As many real-world reasoning scenarios require integrating information across modalities, future extensions should incorporate image–text or video–text benchmarks to more comprehensively assess LLMs' generalization and compositional capabilities.

## C  POTENTIAL SOCIETAL IMPACTS

### C.1  POSITIVE IMPACTS

CryptoX enables a deeper, quantitative understanding of compositional reasoning in LLMs, which can drive the development of more robust and generalizable AI systems. By identifying specific failure modes in subproblem decomposition and inference, CryptoX can guide researchers and engineers toward targeted improvements that enhance AI reliability in high-stakes domains such as scientific discovery, education, and decision support.

### C.2  NEGATIVE IMPACTS

The same cryptographic transformations used by CryptoX to stress-test models could be repurposed to obfuscate malicious content or bypass automated moderation systems, facilitating the spread of misinformation or harmful text. Moreover, revealing detailed failure cases in compositional reasoning may expose vulnerabilities that bad actors could exploit to generate deceptive or adversarial inputs at scale.

# D RULES

## D.1 ENCODING RULE

Since sections 2 and 3 primarily introduce and utilize the emoji shuffle encoding method, this section will present and supplement the other two encoding methods. Content D.1 below shows the encoding rules.

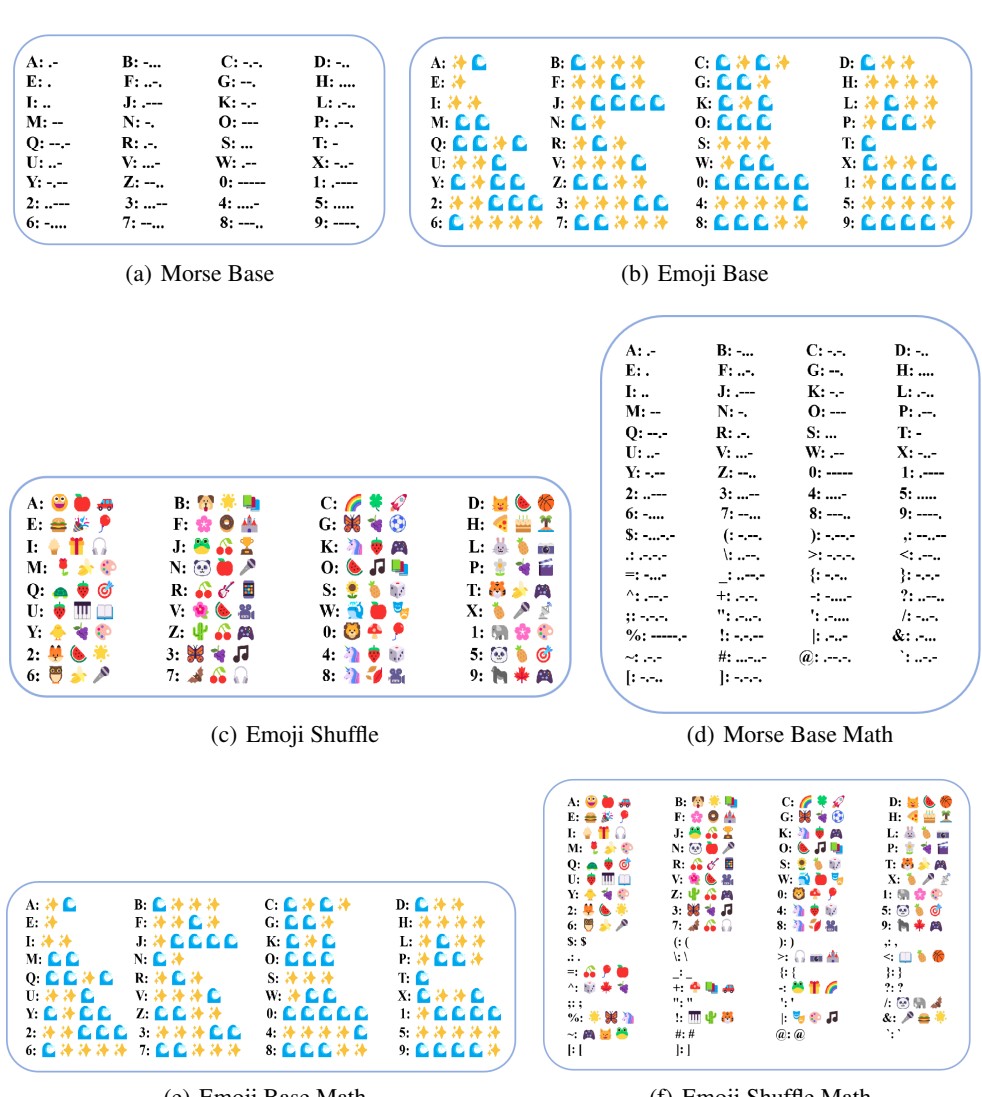

Figure 7: Encoding rule for our experiments.

## D.2 NOISY RULE

The noise addition rule for questions is as follows: A specified number of words are randomly selected, and a random letter is inserted into each selected word at an odd-numbered position. For example, *happy* becomes *hiapnpyt*.

## D.3 DIFFICULT TRANSFORMATION RULE

These difficult transform rules will change the selected word in the following order.

**Duplicate each string**  For example, *happy* becomes *hhaappppyy*.

**Shift each letter one position to the right in the alphabet**  For example, *happy* becomes *ibqqz*.

**Shift a string one position to the right, cyclically**  For example, *happy* becomes *yhapp*.

**Reverse a string**  For example, *happy* becomes *yppah*.

**Shift a string's characters two positions to the left**  For example, *happy* becomes *ppyha*.

**Rotate the letters in even positions**  For example, *happy* becomes *hbpqy*.

**Rotate the letters in odd positions**  For example, *happy* becomes *iaqpz*.

## D.4 LLM AS JUDGER SETTINGS

For the LLM-as-judge, we adopt Doubao-Pro-256K with temperature $T = 0.7$ and top-$p = 0.75$. These settings are carefully optimized through preliminary experiments and consistently yield stable judgments across runs. To further guarantee reliability, we employ fixed prompts, enforce self-consistency checks, and confirm high agreement across repeated evaluations. In addition, a dedicated leakage audit ensures that cipher mappings are not memorized and that answers are not over-credited. Together, these measures provide strong assurance of the robustness and fairness of our evaluation protocol.

# E  ABLATION EXPERIMENTS

Content E below shows the complete results of the experiments, strongly demonstrating that CryptoX reliably captures the relationship between model capacity and compositional reasoning ability.

## E.1  THE EFFECT OF MODEL SIZE

As the model size increases, the accuracy of its responses improves, clearly indicating that the model's compositional reasoning ability is related to its size.

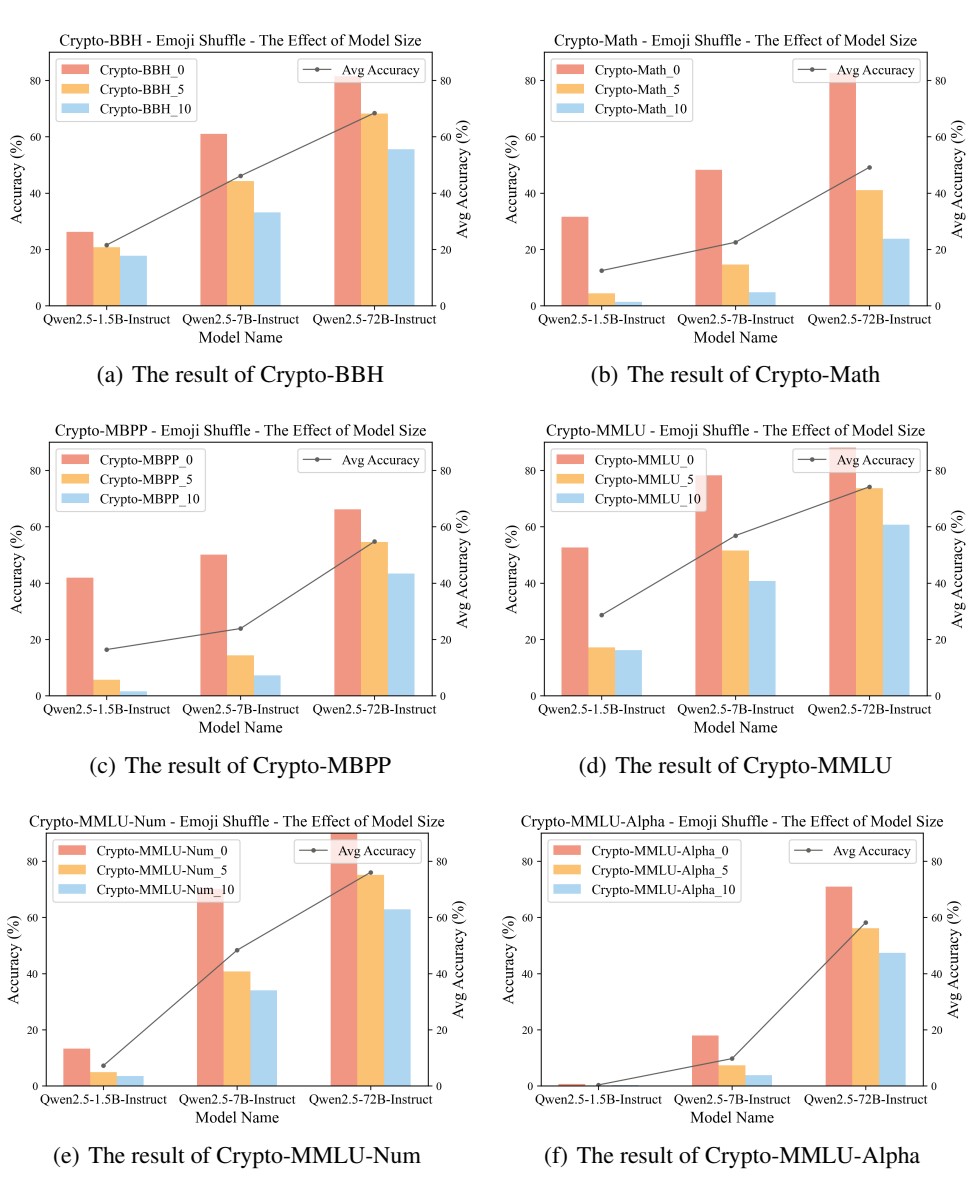

(a) The result of Crypto-BBH

(b) The result of Crypto-Math

(c) The result of Crypto-MBPP

(d) The result of Crypto-MMLU

(e) The result of Crypto-MMLU-Num

(f) The result of Crypto-MMLU-Alpha

Figure 8: The performance of models with different model size. In *Domain_Words*, *Words* denotes the number of words encoded in the given question.

## E.2   THE PERFORMANCE OF DOUBAO-MOE AND DOUBAO-DENSE

Compared to Doubao-Dense, Doubao-Moe performs better on our CryptoBench,clearly indicating that MOE has superior CR capabilities.

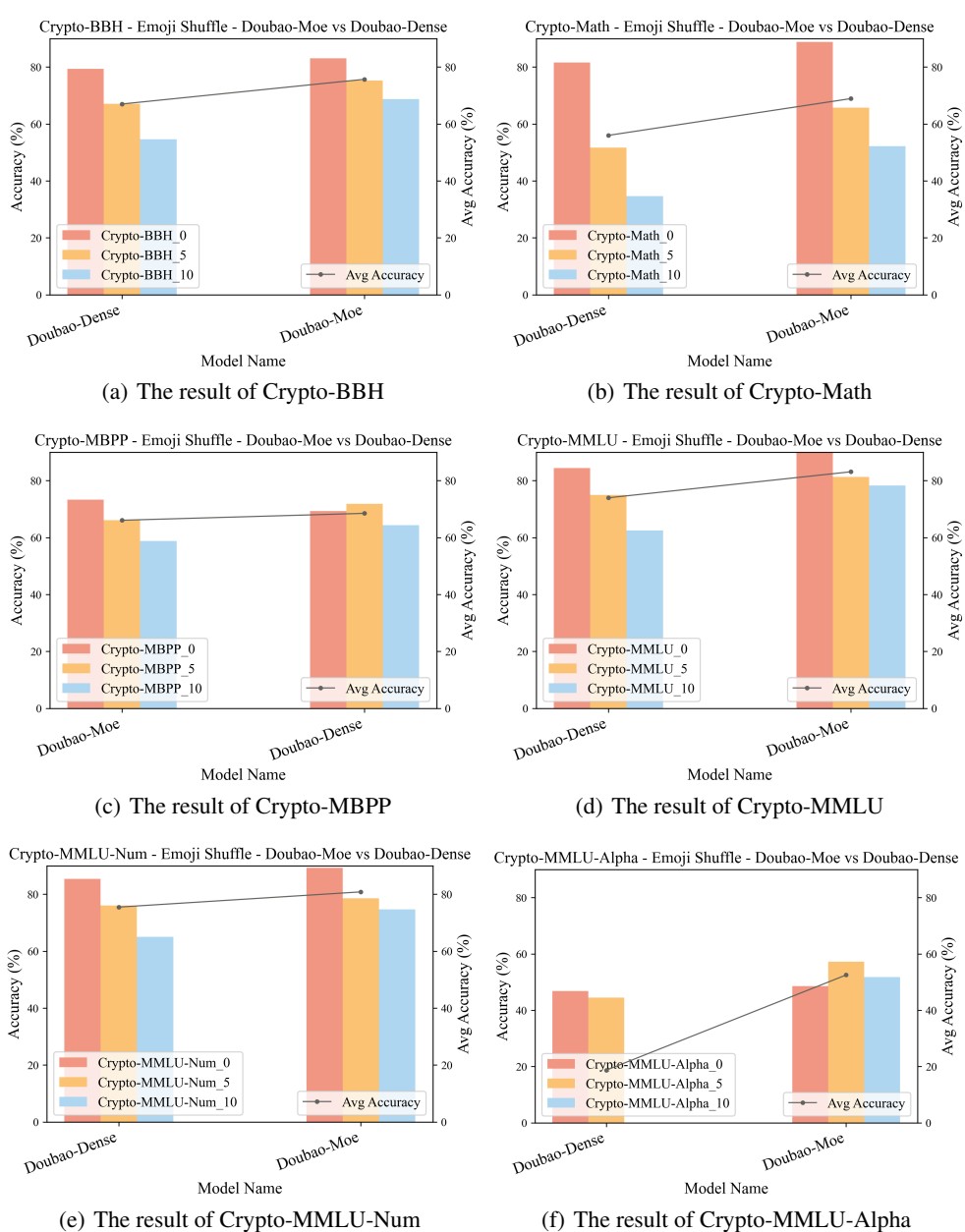

Figure 9: The performance of Doubao-Moe and Doubao-Dense. In *Domain_Words*, *Words* denotes the number of words encoded in the given question.

### E.3 THE COMPARISON OF MULTI-TURN AND SINGLE-TURN DIALOGUE EFFECTS

**Experiment Setup**

- **Multi-Turn**: (1) Complete the task of decoding the encoded question. (2) Answer the decoded question.
- **Single-Turn**: Complete the decoding and answering of the question under the given encoding rules.

**The Performance of Multi-Turn and Single-Turn Dialogue Effects**  The performance of multi-turn dialogues is generally better than that of single-turn dialogues, providing comprehensive evidence that our CryptoBench can effectively validate the model's compositional reasoning ability.

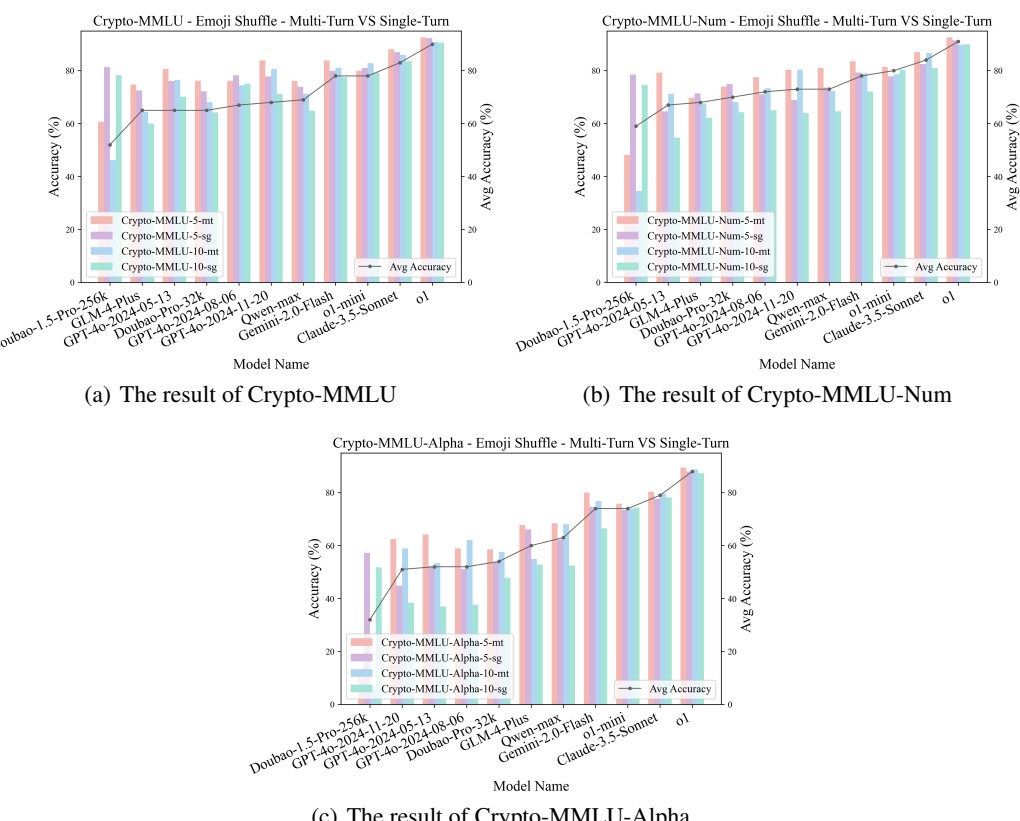

(a) The result of Crypto-MMLU      (b) The result of Crypto-MMLU-Num

(c) The result of Crypto-MMLU-Alpha

Figure 10: The performance of models with different model size. In *Domain_Words*, *Words* denotes the number of words encoded in the given question.

### E.4 THE PERFORMANCE OF BASE AND INSTRUCT LLMS

The results show that instruct models consistently achieve higher performance than base models, providing clear evidence that instruction fine-tuning effectively enhances a model's compositional reasoning ability.

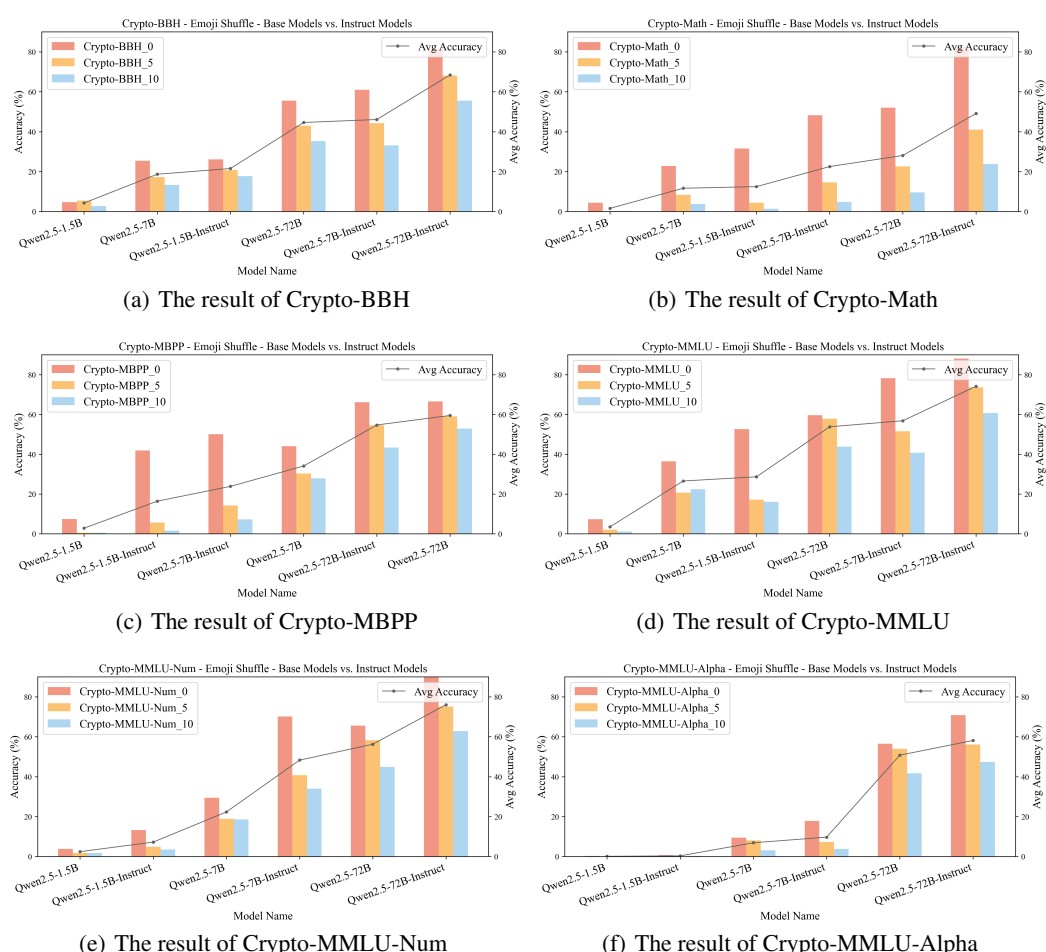

(a) The result of Crypto-BBH

(b) The result of Crypto-Math

(c) The result of Crypto-MBPP

(d) The result of Crypto-MMLU

(e) The result of Crypto-MMLU-Num

(f) The result of Crypto-MMLU-Alpha

Figure 11: The performance between base models and instruct models. In *Domain_Words*, *Words* denotes the number of words encoded in the given question.

## E.5 THE EFFECT OF DIFFERENT ARCHITECTURES

The observation that models with non-mainstream architectures perform worse than smaller models such as Qwen2.5 and Llama-3.1 provides clear evidence that model architecture plays a critical role in shaping compositional reasoning ability.

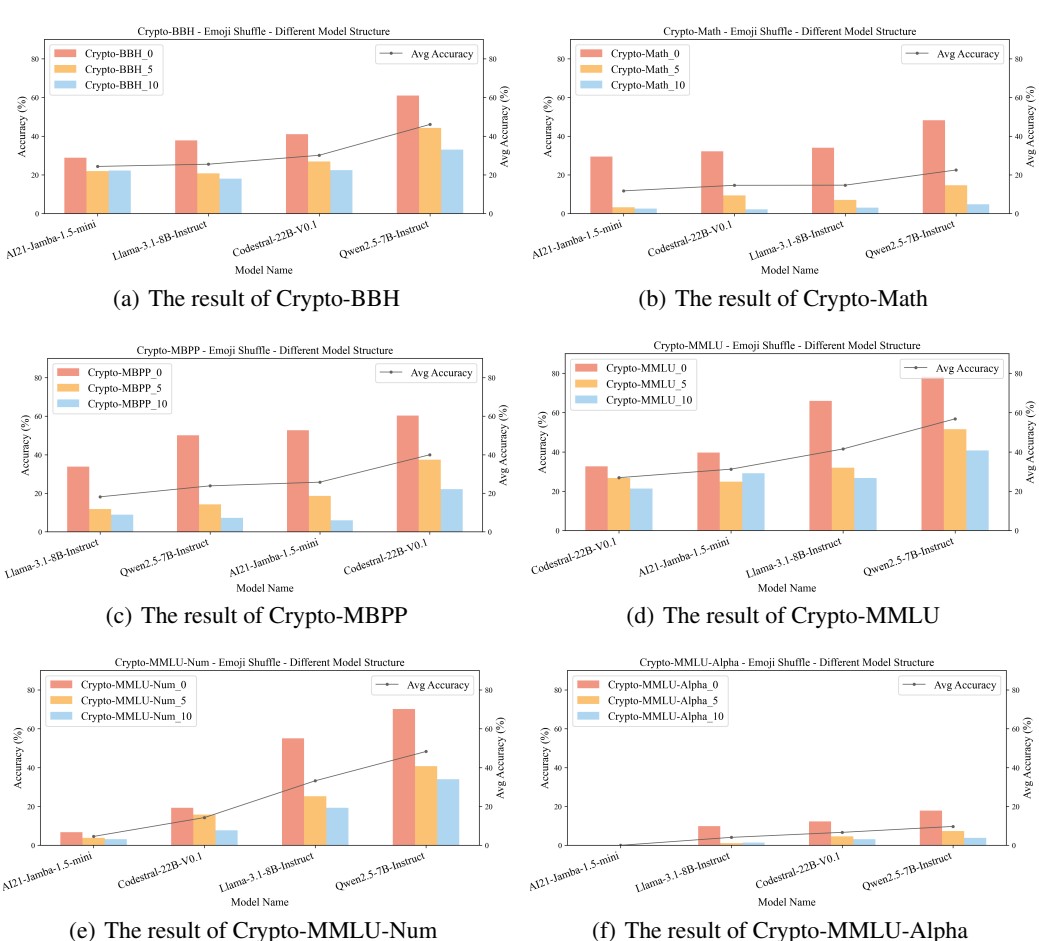

(a) The result of Crypto-BBH

(b) The result of Crypto-Math

(c) The result of Crypto-MBPP

(d) The result of Crypto-MMLU

(e) The result of Crypto-MMLU-Num

(f) The result of Crypto-MMLU-Alpha

Figure 12: The performance of different architectural models. In *Domain_Words*, *Words* denotes the number of words encoded in the given question.

### E.6 THE DECODING CAPACITY OF LLMS

Content E.6 presents the decoding performance of different models, providing strong complementary evidence for the multi-turn vs. single-turn experiments and for the effectiveness of CryptoX.

**ROUGE-1 Score** The statistical results of decoding accuracy using ROUGE-1(22) are presented below, providing clear evidence of the reliability and robustness of our evaluation.

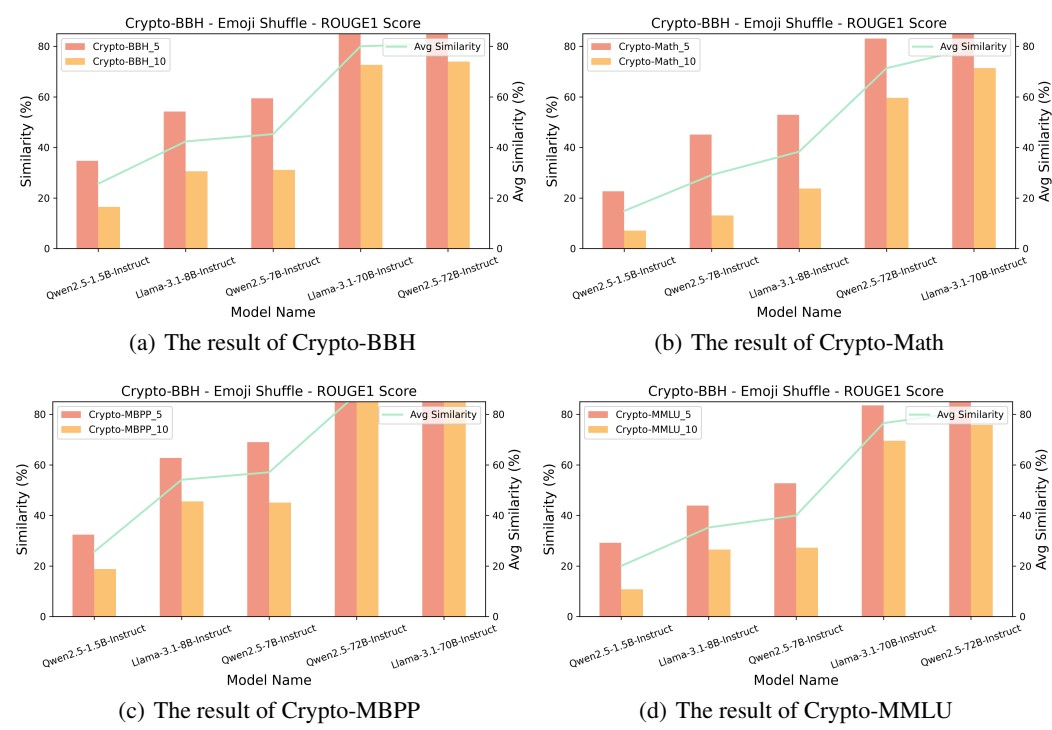

(a) The result of Crypto-BBH  (b) The result of Crypto-Math

(c) The result of Crypto-MBPP  (d) The result of Crypto-MMLU

Figure 13: The decoding performance of different models. In *Domain_Words*, *Words* denotes the number of words encoded in the given question.

**BLEU Score**   The statistical results of decoding accuracy using BLEU(1-gram)(31) are shown below, providing clear evidence of the reliability and robustness of our evaluation.

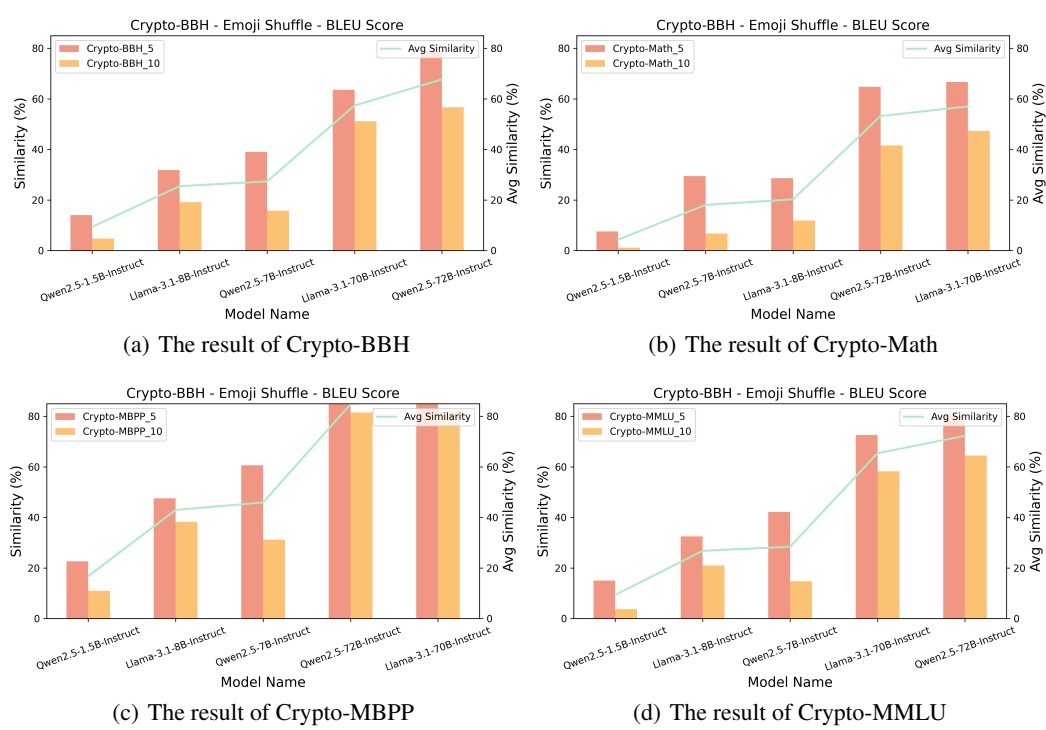

(a) The result of Crypto-BBH

(b) The result of Crypto-Math

(c) The result of Crypto-MBPP

(d) The result of Crypto-MMLU

Figure 14: The decoding performance of different models. In *Domain_Words*, *Words* denotes the number of words encoded in the given question.

## E.7 SFT Cannot Overcome Model Fundamentals in CryptoBench

The following results demonstrate that even after Supervised Fine-Tuning (SFT) of the base model (The data for SFT is sourced from Crypto-MMLU), its performance remains inferior to top-tier counterparts (SFT model achieves an AUC of 2.16 versus o1's 4.05). This indicates that intrinsic model capabilities – particularly fundamental parameters like model scale and architectural design – impose critical constraints on practical effectiveness.This also provides strong evidence that CryptoX remains robust, as its evaluation outcomes are not artificially inflated by SFT, thereby reliably reflecting models' intrinsic compositional reasoning ability.

Table 6: Comparison of Model Performance After Supervised Fine-Tuning

| Model | AUC | Avg | Crypto-Math | | | Crypto-MBPP | | | Crypto-BBH | | | Crypto-MMLU | | | Crypto-MMLU-Num | | | Crypto-MMLU-Alpha | | | Crypto-Needle-30K | | |
|---|---|---|---|---|---|---|---|---|---|---|---|---|---|---|---|---|---|---|---|---|---|---|---|
| | | | 0 | 5 | 10 | 0 | 5 | 10 | 0 | 5 | 10 | 0 | 5 | 10 | 0 | 5 | 10 | 0 | 5 | 10 | 0 | 5 | 10 |
| o1 | 4.05 | 83.69 | 96.99 | 89.66 | 84.48 | 64.93 | 68.04 | 63.9 | 84.08 | 83.66 | 82.13 | 94.35 | 92.25 | 90.53 | 92.5 | 91.43 | 90.0 | 90.07 | 87.99 | 87.32 | 99.66 | 80.33 | 43.2 |
| o3-mini | 3.67 | 76.38 | 95.99 | 85.4 | 77.62 | 46.96 | 60.39 | 57.49 | 80.6 | 72.03 | 72.7 | 88.34 | 86.97 | 84.91 | 88.93 | 88.57 | 86.07 | 87.23 | 85.87 | 83.1 | 88.55 | 52.0 | 34.35 |
| SFT | 2.16 | 49.98 | 73.55 | 44.42 | 27.62 | 54.32 | 56.28 | 50.46 | 71.89 | 63.37 | 54.34 | 84.1 | 82.75 | 78.25 | 70.71 | 70.71 | 65.36 | 11.35 | 5.65 | 4.58 | 78.79 | 0.67 | 0.34 |
| Base | 1.4 | 40.21 | 75.75 | 30.43 | 13.11 | 58.38 | 40.03 | 23.8 | 69.15 | 55.45 | 47.39 | 86.22 | 63.38 | 50.53 | 66.07 | 27.14 | 23.21 | 17.02 | 9.54 | 5.63 | 81.14 | 1.0 | 0.0 |

## E.8 Variance of Accuracy and AUC for Different Models

It can be observed that although the variance increases with rising difficulty, it remains consistently much lower than the variance associated with Accuracy, thereby demonstrating the effectiveness of the AUC metric. In our experiments, model scores almost always decrease monotonically with increasing difficulty, making "triangular" variations highly unlikely in practice. To further ensure validity, we incorporate length-matched and random-glyph controls, as well as per-difficulty normalization, which confirm that AUC isolates compositional reasoning rather than artifacts of instruction length, spacing, or output control. In practice, we recommend interpreting AUC alongside per-difficulty scores; when overall accuracies are similar, AUC provides substantially better discriminatory power.

Table 7: Variance of accuracy and AUC for closed-source models on Crypto-HighResolution. *Num* stands for Accuracy, which corresponds to solving the question of *Num* words being encoded.

| | 0 | 1 | 2 | 3 | 4 | 5 | 6 | 7 | 8 | 9 | 10 | AUC |
|---|---|---|---|---|---|---|---|---|---|---|---|---|
| **Variance** | 0.0043 | 0.0055 | 0.0058 | 0.0058 | 0.0066 | 0.0069 | 0.0081 | 0.0094 | 0.0091 | 0.0116 | 0.0137 | **0.6221** |

### E.9 CORRELATION WITH CHATBOT ARENA

Chatbot Arena (5), which builds its evaluation system through human interaction, is regarded as the "gold standard" in the assessment of human-computer dialogue systems. Although this method has some biases (such as differences in user background and insufficient task diversity), its large scale and high ecological validity make it one of the most representative human evaluation frameworks.

To examine the alignment between our approach and human evaluation systems, we employ Spearman correlation to analyze the relationship among CryptoBench, MMLU, and Chatbot Arena rankings. Spearman correlation is particularly well-suited for nonlinear distributions and reliably captures monotonic relationships, making it a widely adopted method for studying the connection between evaluation metrics and human preferences. This ensures strong statistical support for our findings.

As shown in Figure 15, both AUC and Avg scores from CryptoBench exhibit substantially higher correlations with human preferences (0.61 and 0.57, respectively) compared to MMLU (0.19). These results provide compelling evidence that CryptoBench serves as a more faithful proxy for human judgment of compositional reasoning ability.

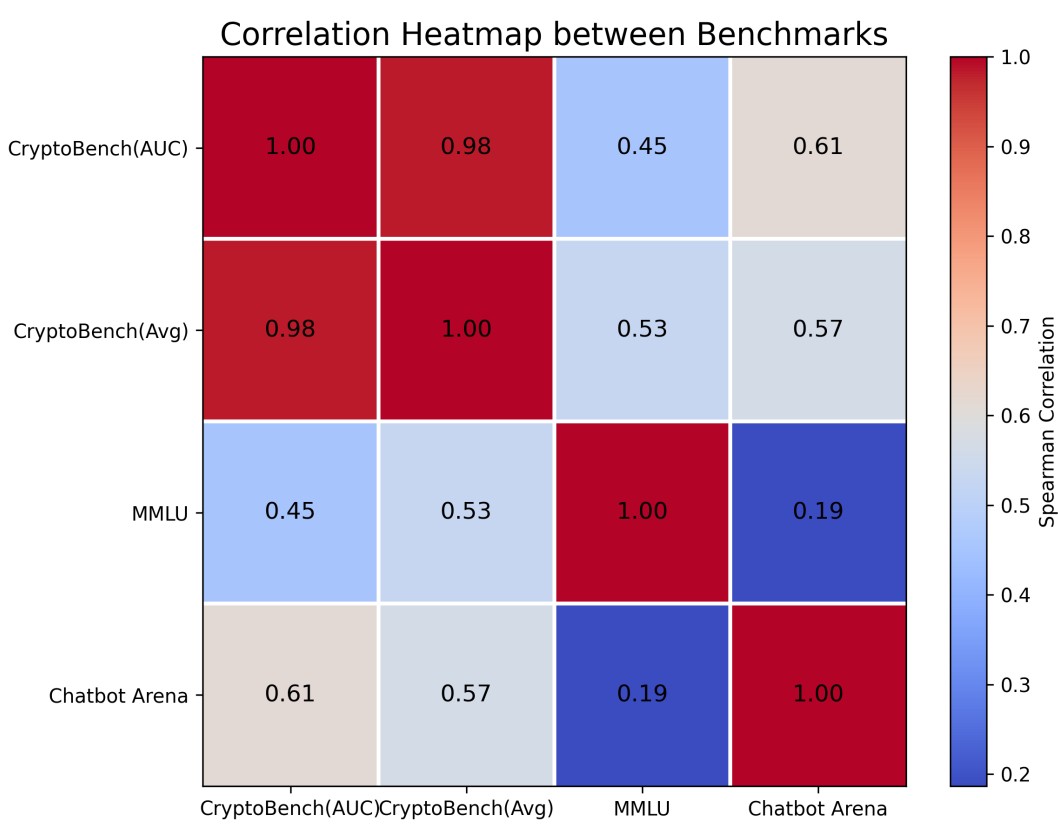

Figure 15: The correlation matrix for benchmarks. The closer the Spearman correlation is to 1, the more similar the rankings of the two benchmarks are.

### E.10 CORRELATION WITH SIMILAR BENCHMARKS

The results demonstrate that while scores on CryptoBench correlate with those on general reasoning benchmarks, CryptoBench also preserves its unique evaluation perspective. This dual property establishes a strong connection between CryptoBench performance and broader compositional reasoning abilities.

To validate this, we evaluate 18 models (including o1, Claude, Gemini, and DeepSeek) on KOR-Bench (24), ZebraLogic (3), and CryptoBench, and analyze their rank correlations (ranging from –1 to 1, with higher values indicating stronger alignment).

As shown in Figure 16, model performance on CryptoBench exhibits significant correlations with both KORBench and ZebraLogic. These findings provide compelling evidence that CryptoBench effectively captures compositional reasoning ability while maintaining distinctive evaluative strength.

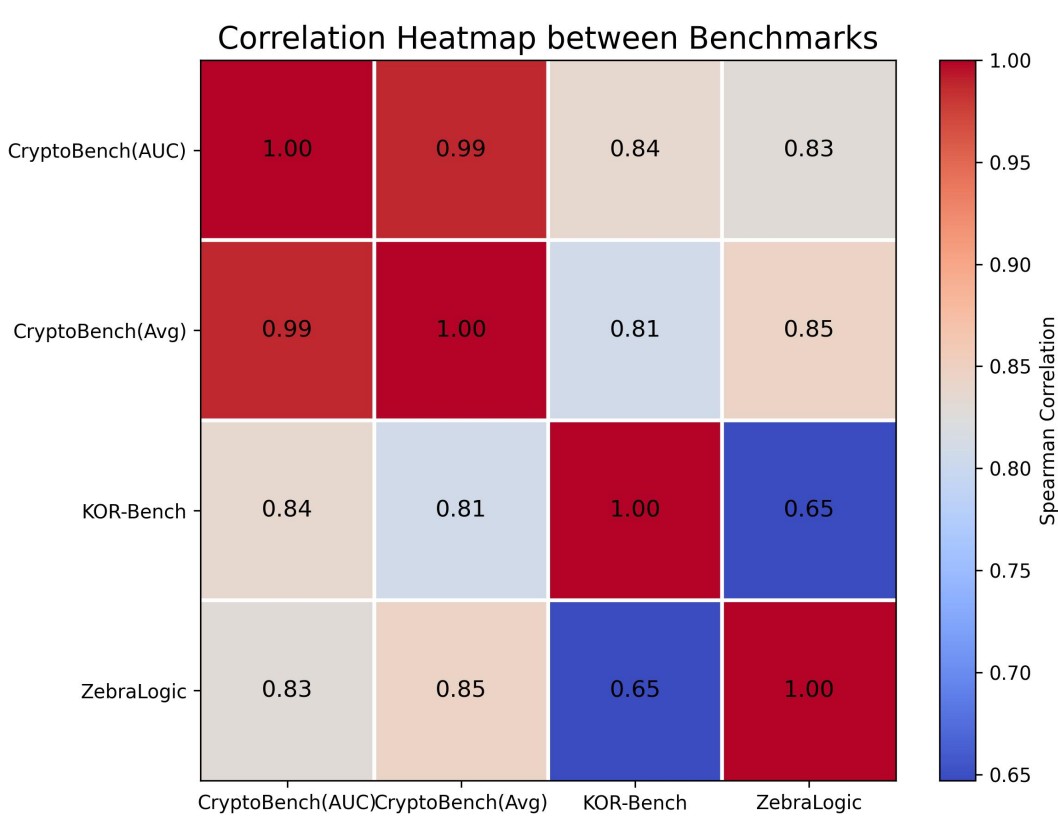

Figure 16: The correlation matrix for benchmarks. The closer the Spearman correlation is to 1, the more similar the rankings of the two benchmarks are.

## E.11 RULE-FREE EXAMPLE-BASED MODEL PERFORMANCE

The experimental design adopts 3-shot prompts, both with and without explicit encoding rules, where the three exemplars collectively cover the English alphabet. The results show that most models achieve comparable accuracy in both settings, indicating that they can autonomously infer encoding patterns directly from the given examples. This provides strong evidence of models' inherent capacity to internalize transformation rules without explicit guidance. The detailed results are presented below:

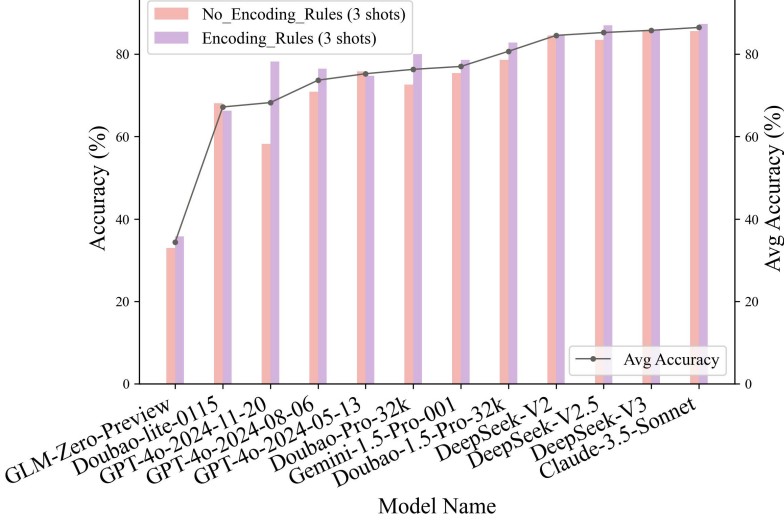

Figure 17: Comparison of the Presence and Absence of Encoding Rules in Prompt.

## E.12 MODEL PERFORMANCE ON THE NOISY RULE

In this experiment, words are encoded using the emoji shuffle rule, even after the introduction of noise. By jointly incorporating the challenges of noise and encoding, this setting provides a rigorous evaluation of models' compositional reasoning abilities. The results show that Claude-3.7-Sonnet-Thinking achieves the best performance, closely followed by Gemini-2.5-Pro-Preview-0506. Notably, this ranking differs from that observed on Chatbot Arena, providing strong evidence that our constructed compositional tasks reveal nuanced limitations of Gemini-2.5-Pro-Preview-0506 and highlight the effectiveness of our evaluation framework in distinguishing model capabilities.

Table 8: Performance of different models on noisy questions. 5 / 10 represents the number of words encoded. green represents the number with the highest accuracy, blue represents the number with the second highest accuracy, and orange represents the value with the third highest accuracy.

| Model | AUC | Avg | Crypto-Math | | Crypto-MBPP | | Crypto-BBH | | Crypto-MMLU | |
| --- | --- | --- | --- | --- | --- | --- | --- | --- | --- | --- |
| | | | 5 | 10 | 5 | 10 | 5 | 10 | 5 | 10 |
| Claude-3.7-Sonnet-Thinking | 3.61 | 72.25 | 67.28 | 58.86 | 88.54 | 89.83 | 76.54 | 72.59 | 64.21 | 60.18 |
| Claude-3.7-Sonnet | 3.42 | 68.4 | 53.6 | 38.6 | 85.89 | 86.48 | 57.41 | 52.96 | 87.37 | 84.91 |
| DeepSeek-R1 | 3.39 | 67.87 | 56.6 | 34.8 | 84.86 | 83.28 | 65.31 | 59.14 | 84.56 | 74.39 |
| o4-mini-0416 | 3.37 | 67.35 | 66.8 | 47.0 | 80.05 | 78.59 | 61.98 | 54.94 | 78.6 | 70.88 |
| Gemini-2.5-Pro-Preview-0506 | 3.29 | 65.83 | 67.2 | 53.4 | 72.29 | 63.39 | 51.61 | 48.27 | 89.47 | 81.05 |
| Claude-3.5-Sonnet | 3.28 | 65.61 | 42.0 | 29.8 | 73.37 | 73.31 | 76.05 | 69.14 | 82.81 | 78.42 |
| o1-1217 | 3.19 | 63.86 | 60.2 | 40.8 | 85.64 | 87.43 | 39.38 | 33.95 | 86.67 | 76.84 |
| DeepSeek-V3-0324 | 3.14 | 62.81 | 49.4 | 33.2 | 78.91 | 74.85 | 61.23 | 53.7 | 81.75 | 69.47 |
| Gemini-2.5-Flash-Preview-0417 | 2.93 | 58.6 | 58.6 | 40.2 | 77.09 | 77.67 | 33.83 | 29.51 | 81.4 | 70.53 |
| Gemini-2.0-Flash.001 | 2.83 | 56.52 | 47.8 | 27.6 | 79.39 | 71.24 | 46.91 | 36.42 | 77.19 | 65.61 |
| GPT4.1-0414 | 2.79 | 55.87 | 48.4 | 33.6 | 84.93 | 83.04 | 18.27 | 15.19 | 87.72 | 75.79 |
| GPT4.1-mini-0414 | 2.7 | 53.91 | 51.0 | 29.4 | 63.14 | 56.91 | 45.56 | 40.37 | 77.9 | 67.02 |
| Gemma3-27b | 2.32 | 46.42 | 34.6 | 14.2 | 65.09 | 62.6 | 37.9 | 35.56 | 65.26 | 56.14 |
| GPT4o-2024-08-06 | 2.27 | 45.45 | 31.8 | 15.8 | 71.0 | 68.29 | 22.84 | 18.77 | 72.28 | 62.81 |
| Qwenmax | 2.11 | 42.26 | 32.0 | 17.6 | 66.04 | 57.79 | 19.51 | 15.68 | 69.47 | 60.0 |
| GPT4o-mini-0718 | 2.08 | 41.64 | 20.0 | 9.4 | 65.43 | 61.33 | 42.59 | 33.7 | 54.03 | 46.67 |
| GPT4o-2024-11-20 | 1.94 | 38.8 | 29.8 | 13.2 | 65.09 | 54.74 | 21.23 | 16.91 | 61.05 | 48.42 |
| GPT4.1-nano-0414 | 1.94 | 38.82 | 25.6 | 12.0 | 55.22 | 49.79 | 35.93 | 26.42 | 57.54 | 48.07 |

### E.13 MODEL PERFORMANCE ON THE DIFFICULT TRANSFORMATION RULE

In this experiment, we design a more intricate transformation setting in which selected words undergo seven distinct transformations sequentially, followed by the application of our existing word encoding rule. This setup demonstrates that our proposed CryptoX is not limited to a single transformation mode such as word encoding, but can also robustly accommodate more complex and diverse transformation processes. Remarkably, when handling these challenging transformation tasks, DeepSeek-R1 achieves the best performance, despite ranking lower on Chatbot Arena. This divergence provides strong evidence that CryptoX offers a more sensitive and effective measure of models' compositional reasoning capabilities.

Table 9: Performance of different models on difficult transformation questions. 5 / 10 represents the number of words encoded. green represents the number with the highest accuracy, blue represents the number with the second highest accuracy, and orange represents the value with the third highest accuracy.

| Model | AUC | Avg | Crypto-Math | | Crypto-MBPP | | Crypto-BBH | | Crypto-MMLU | |
|---|---|---|---|---|---|---|---|---|---|---|
| | | | 5 | 10 | 5 | 10 | 5 | 10 | 5 | 10 |
| DeepSeek-R1 | 3.13 | 62.52 | 43.6 | 26.0 | 85.48 | 85.25 | 59.01 | 50.99 | 80.0 | 69.83 |
| o4-mini-0416 | 2.95 | 59.02 | 49.2 | 28.8 | 79.29 | 76.81 | 54.2 | 44.94 | 74.03 | 64.91 |
| DeepSeek-V3-0324 | 2.91 | 58.19 | 35.6 | 19.6 | 79.2 | 76.57 | 57.28 | 52.35 | 78.25 | 66.67 |
| o1-1217 | 2.9 | 58.04 | 48.0 | 26.2 | 86.73 | 85.01 | 37.04 | 30.12 | 80.7 | 70.53 |
| Claude-3.7-Sonnet | 2.77 | 55.46 | 35.8 | 20.2 | 85.65 | 83.47 | 42.96 | 29.63 | 76.84 | 69.12 |
| Claude-3.5-Sonnet | 2.74 | 54.78 | 31.4 | 17.6 | 70.67 | 66.38 | 60.37 | 47.28 | 75.44 | 69.12 |
| Gemini-2.5-Pro-Preview-0506 | 2.68 | 53.59 | 48.4 | 28.4 | 64.33 | 55.04 | 42.47 | 38.15 | 81.05 | 70.88 |
| GPT4.1-mini-0414 | 2.52 | 50.42 | 38.0 | 21.0 | 65.91 | 64.28 | 44.2 | 34.2 | 71.93 | 63.86 |
| Gemini-2.0-Flash.001 | 2.51 | 50.28 | 36.2 | 18.2 | 72.49 | 63.58 | 41.23 | 31.61 | 75.79 | 63.16 |
| Gemini-2.5-Flash-Preview-0417 | 2.49 | 49.71 | 43.8 | 23.6 | 78.34 | 75.37 | 27.04 | 20.37 | 70.17 | 58.95 |
| GPT4.1-0414 | 2.48 | 49.51 | 37.2 | 19.2 | 81.87 | 79.7 | 16.05 | 12.96 | 78.95 | 70.17 |
| GPT4o-2024-08-06 | 2.05 | 41.03 | 22.6 | 10.8 | 70.78 | 64.87 | 21.11 | 18.39 | 64.21 | 55.44 |
| GPT4o-mini-0718 | 1.94 | 38.84 | 17.2 | 7.6 | 64.36 | 59.94 | 39.01 | 29.63 | 49.47 | 43.51 |
| Qwenmax | 1.94 | 38.88 | 21.2 | 11.4 | 68.16 | 60.28 | 20.12 | 15.19 | 61.05 | 53.68 |
| GPT4o-2024-11-20 | 1.81 | 36.14 | 17.6 | 9.0 | 60.66 | 58.0 | 23.58 | 16.42 | 59.3 | 44.56 |
| GPT4.1-nano-0414 | 1.74 | 34.86 | 21.8 | 8.4 | 49.45 | 39.36 | 37.16 | 27.65 | 52.98 | 42.1 |

# F LOGIT LENS ANALYSIS(0%/50%/100% ENCODING RATIO)

While Section 4 primarily focuses on the emoji shuffle encoding method with 0/3/5 encoded words, this section extends the analysis by incorporating experiments with 0%/50%/100% encoding ratios. These additional results provide more comprehensive and compelling evidence that further substantiates the conclusions drawn in Section 4.

## F.1 BASE MORSE ENCODING RULE

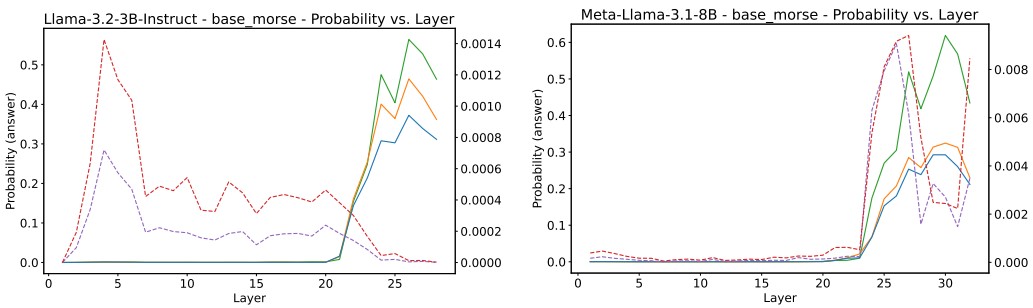

(a) The result of Crypto-MMLU-BaseMorse on Llama-3.2-3B-Instruct
(b) The result of Crypto-MMLU-BaseMorse on Llama-3.1-8B

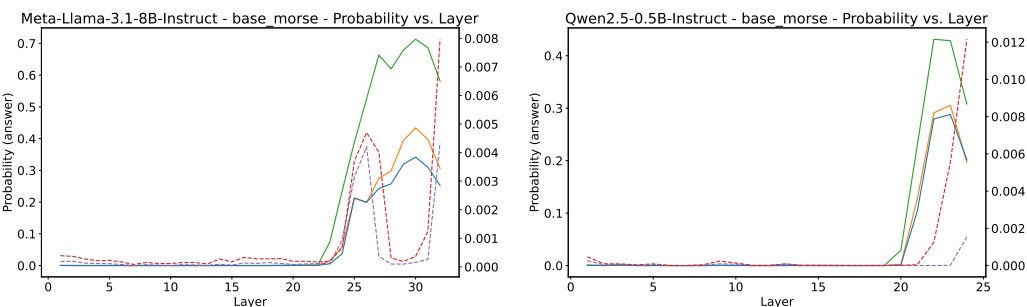

(c) The result of Crypto-MMLU-BaseMorse on Llama-3.1-8B-Instruct
(d) The result of Crypto-MMLU-BaseMorse on Qwen2.5-0.5B-Instruct

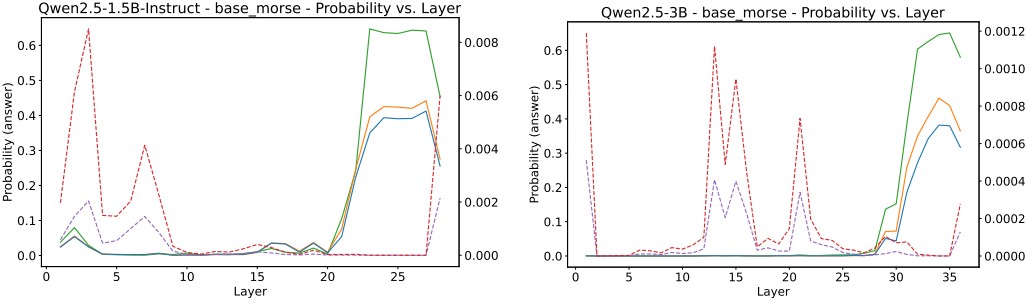

(e) The result of Crypto-MMLU-BaseMorse on Qwen2.5-1.5B-Instruct
(f) The result of Crypto-MMLU-BaseMorse on Qwen2.5-3B

Figure 18: The logit lens analysis on Crypto-MMLU-BaseMorse using 0%/50%/100% encoding ratios.

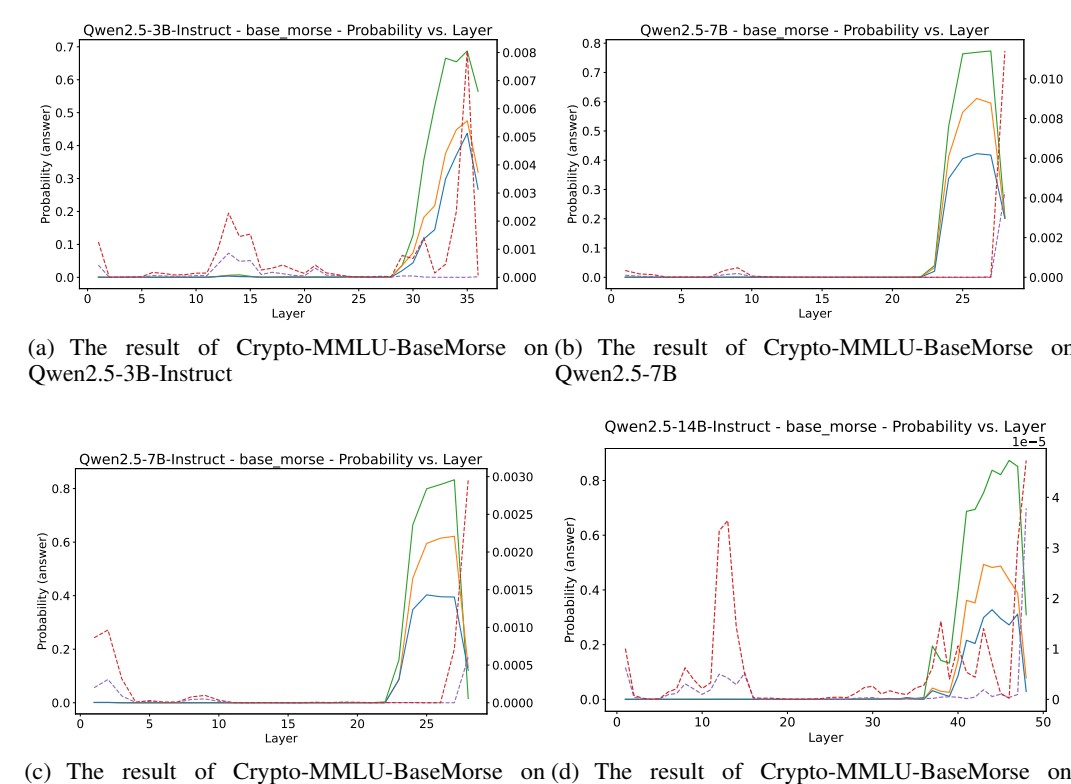

(a) The result of Crypto-MMLU-BaseMorse on Qwen2.5-3B-Instruct

(b) The result of Crypto-MMLU-BaseMorse on Qwen2.5-7B

(c) The result of Crypto-MMLU-BaseMorse on Qwen2.5-7B-Instruct

(d) The result of Crypto-MMLU-BaseMorse on Qwen2.5-14B-Instruct

Figure 19: The logit lens analysis on Crypto-MMLU-BaseMorse using 0%/50%/100% encoding ratios.

## F.2 EMOJI MORSE ENCODING RULE

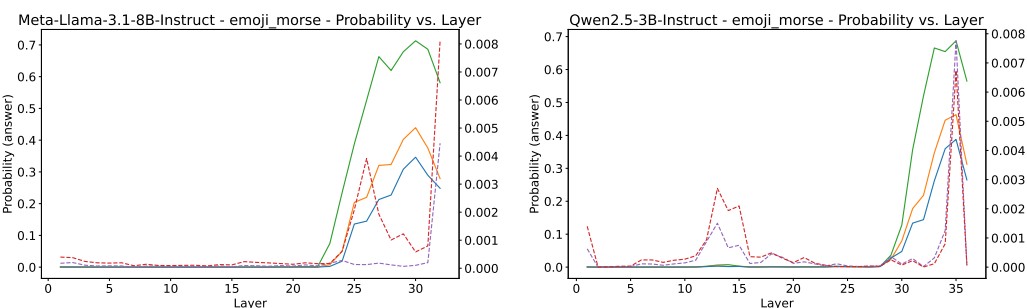

(a) The result of Crypto-MMLU-EmojiMorse on Llama-3.2-3B-Instruct

(b) The result of Crypto-MMLU-EmojiMorse on Llama-3.1-8B

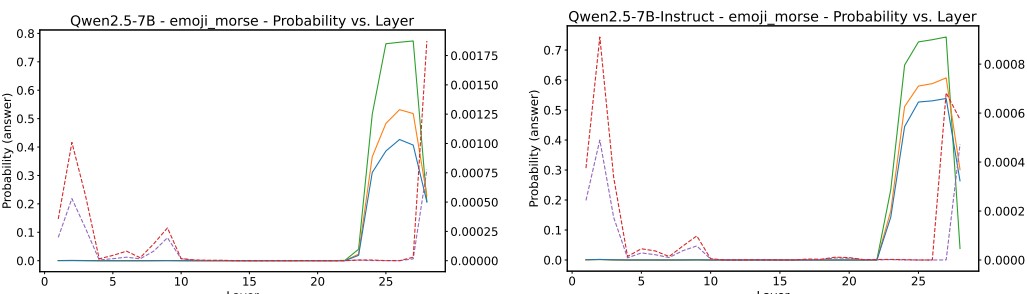

(c) The result of Crypto-MMLU-EmojiMorse on Llama-3.1-8B-Instruct

(d) The result of Crypto-MMLU-EmojiMorse on Qwen2.5-3B-Instruct

(e) The result of Crypto-MMLU-EmojiMorse on Qwen2.5-7B

(f) The result of Crypto-MMLU-EmojiMorse on Qwen2.5-7B-Instruct

Figure 20: The logit lens analysis on Crypto-MMLU-EmojiMorse using 0%/50%/100% encoding ratios.

## F.3 EMOJI SHUFFLE ENCODING RULE

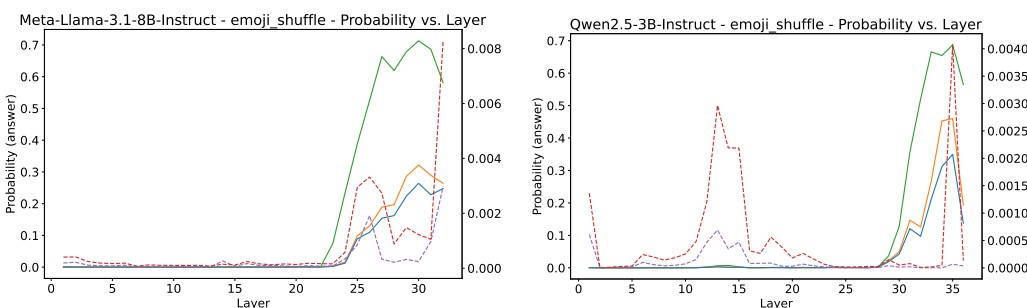

(a) The result of Crypto-MMLU-EmojiShuffle on Llama-3.2-3B-Instruct

(b) The result of Crypto-MMLU-EmojiShuffle on Llama-3.1-8B

(c) The result of Crypto-MMLU-EmojiShuffle on Llama-3.1-8B-Instruct

(d) The result of Crypto-MMLU-EmojiShuffle on Qwen2.5-3B-Instruct

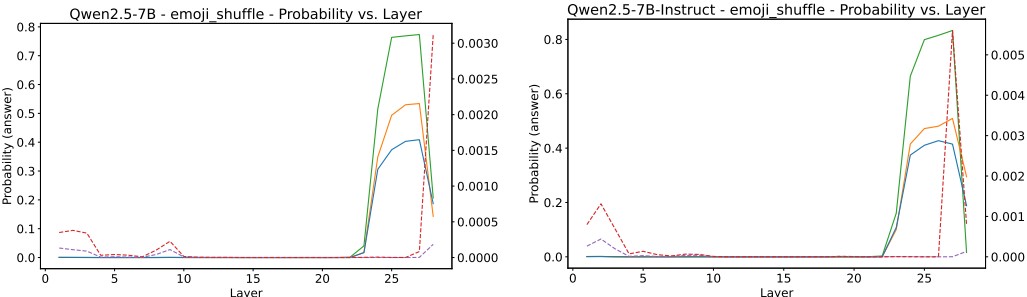

(e) The result of Crypto-MMLU-EmojiShuffle on Qwen2.5-7B

(f) The result of Crypto-MMLU-EmojiShuffle on Qwen2.5-7B-Instruct

Figure 21: The logit lens analysis on Crypto-MMLU-EmojiShuffle using 0%/50%/100% encoding ratios.

# G LOGIT LENS ANALYSIS(0/3/5 ENCODING WORDS)

Since Section 4 primarily employs the emoji shuffle encoding method with 0/3/5 encoded words, this section presents and supplements the complete set of experimental results under the same setting. These additional results offer more comprehensive and compelling evidence that further reinforce the conclusions drawn in Section 4.

## G.1 BASE MORSE ENCODING RULE

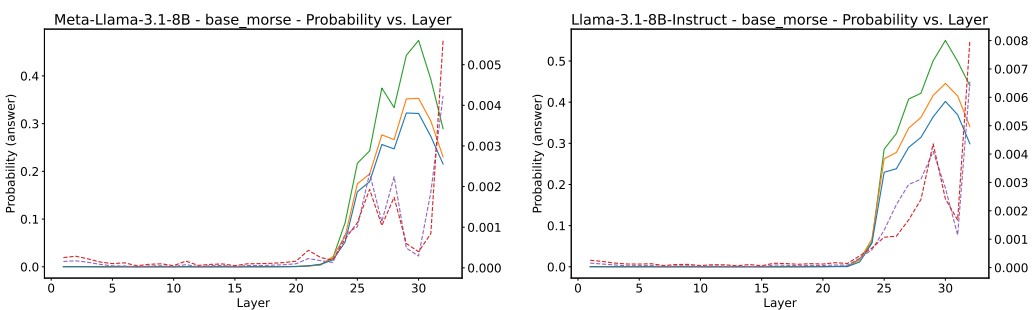

(a) The result of Crypto-MMLU-BaseMorse on Llama-3.1-8B

(b) The result of Crypto-MMLU-BaseMorse on Llama-3.1-8B-Instruct

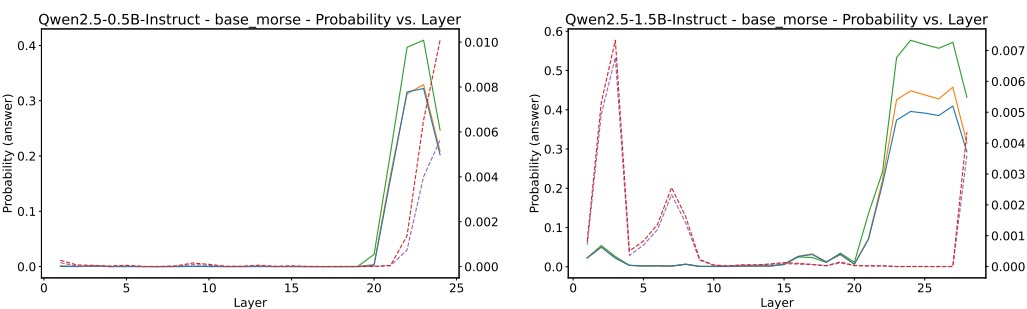

(c) The result of Crypto-MMLU-BaseMorse on Qwen2.5-0.5B-Instruct

(d) The result of Crypto-MMLU-BaseMorse on Qwen2.5-1.5B-Instruct

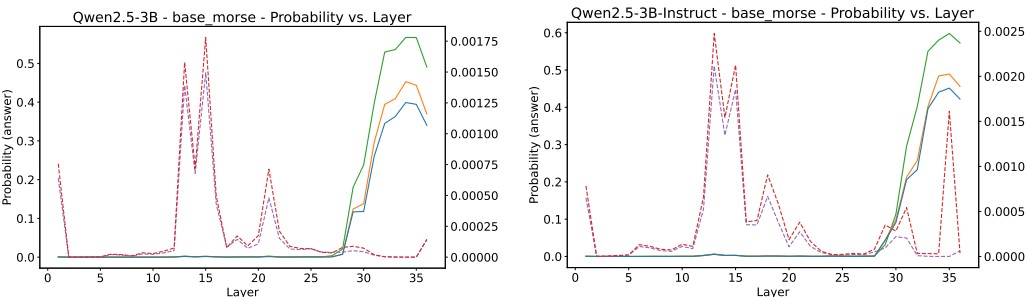

(e) The result of Crypto-MMLU-BaseMorse on Qwen2.5-3B

(f) The result of Crypto-MMLU-BaseMorse on Qwen2.5-3B-Instruct

Figure 22: The logit lens analysis on Crypto-MMLU-BaseMorse using 0/3/5 encoding words.

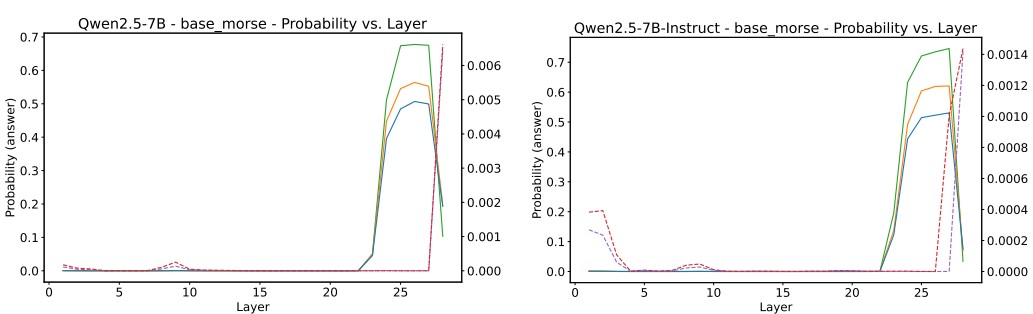

(a) The result of Crypto-MMLU-BaseMorse on Qwen2.5-7B

(b) The result of Crypto-MMLU-BaseMorse on Qwen2.5-7B-Instruct

Figure 23: The logit lens analysis on Crypto-MMLU-BaseMorse using 0/3/5 encoding words.

## G.2 EMOJI MORSE ENCODING RULE

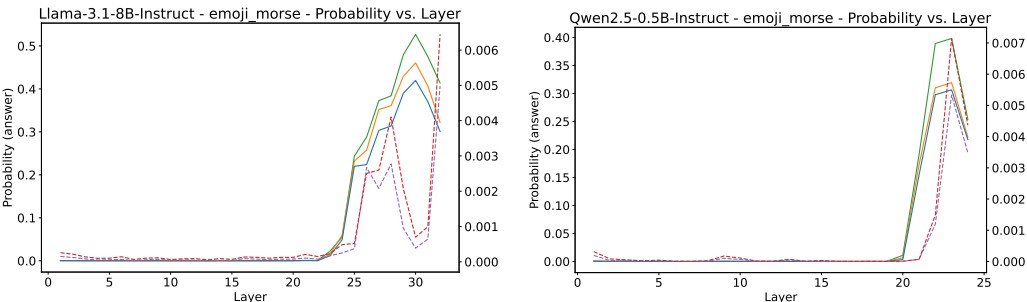

(a) The result of Crypto-MMLU-EmojiMorse on Llama-3.1-8B-Instruct

(b) The result of Crypto-MMLU-EmojiMorse on Qwen2.5-0.5B-Instruct

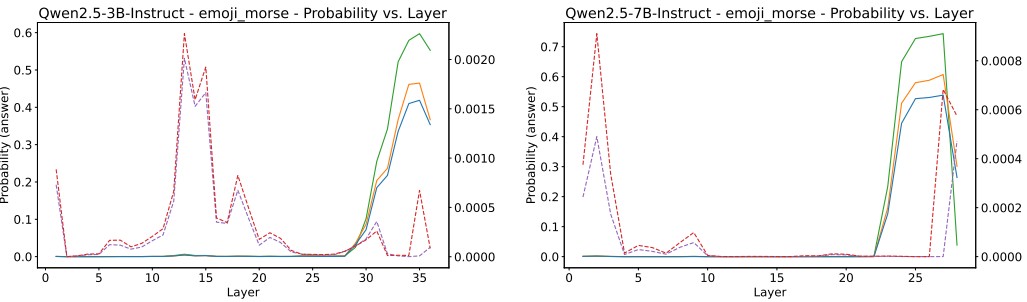

(c) The result of Crypto-MMLU-EmojiMorse on Qwen2.5-3B-Instruct

(d) The result of Crypto-MMLU-EmojiMorse on Qwen2.5-7B-Instruct

Figure 24: The logit lens analysis on Crypto-MMLU-EmojiMorse using 0/3/5 encoding words.

### G.3 EMOJI SHUFFLE ENCODING RULE

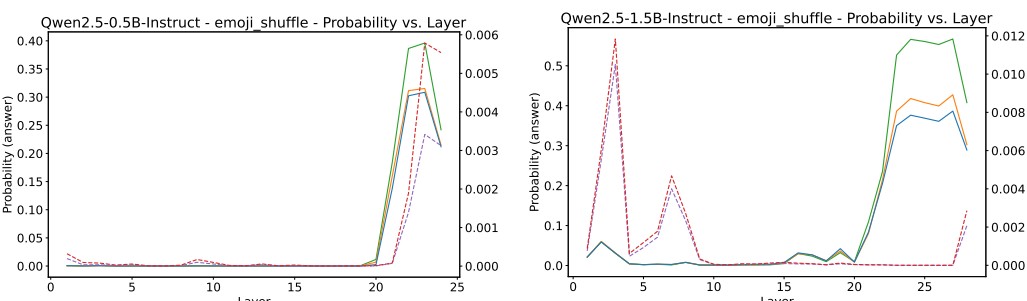

(a) The result of Crypto-MMLU-EmojiShuffle on Qwen2.5-0.5B-Instruct

(b) The result of Crypto-MMLU-EmojiShuffle on Qwen2.5-1.5B-Instruct

Figure 25: The logit lens analysis on Crypto-MMLU-EmojiShuffle using 0/3/5 encoding words.

## H NEURON ACTIVATION ANALYSIS

The appendix H shows result of Neuron Activation Analysis not presented in section 4.

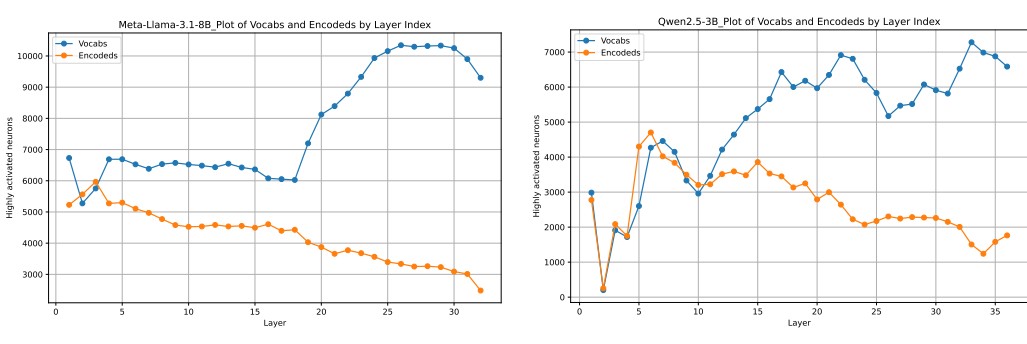

(a) The result of Neuron Activation on Llama-3.1-8B

(b) The result of Neuron Activation on Qwen2.5-3B

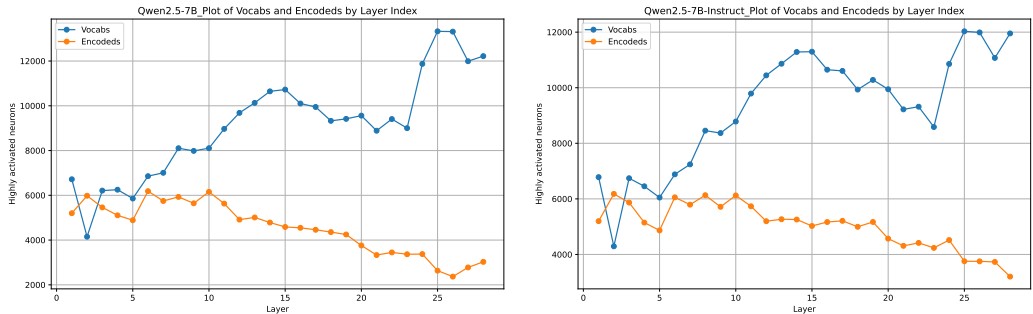

(c) The result of Neuron Activation on Qwen2.5-7B

(d) The result of Neuron Activation on Qwen2.5-7B-Instruct

Figure 26: The other result of Neuron Activation Analysis.

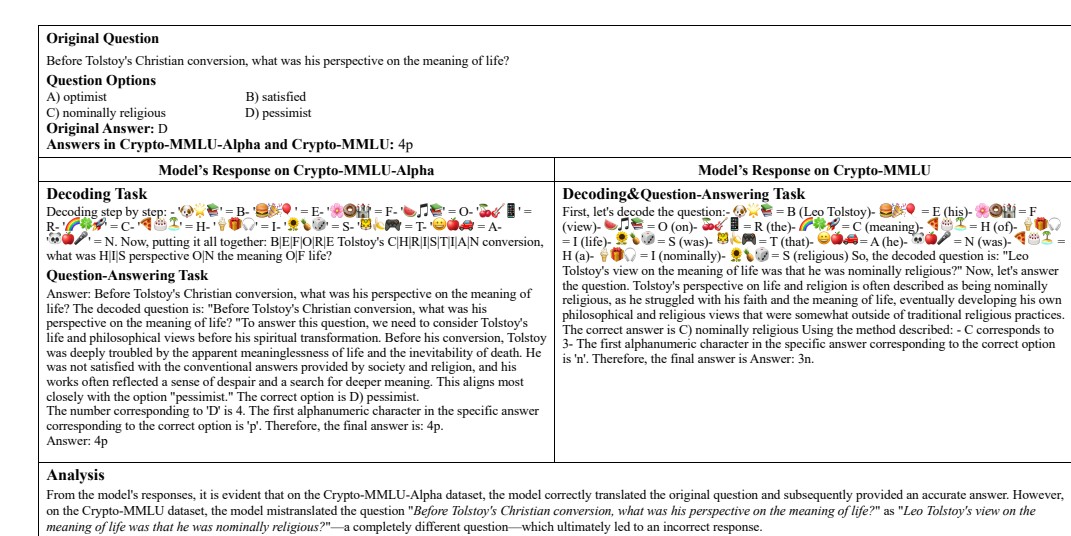

Figure 27: Case 1 in the Case Study: The prompts for the Crypto-MMLU-Alpha and Crypto-MMLU datasets are provided in Appendix J.

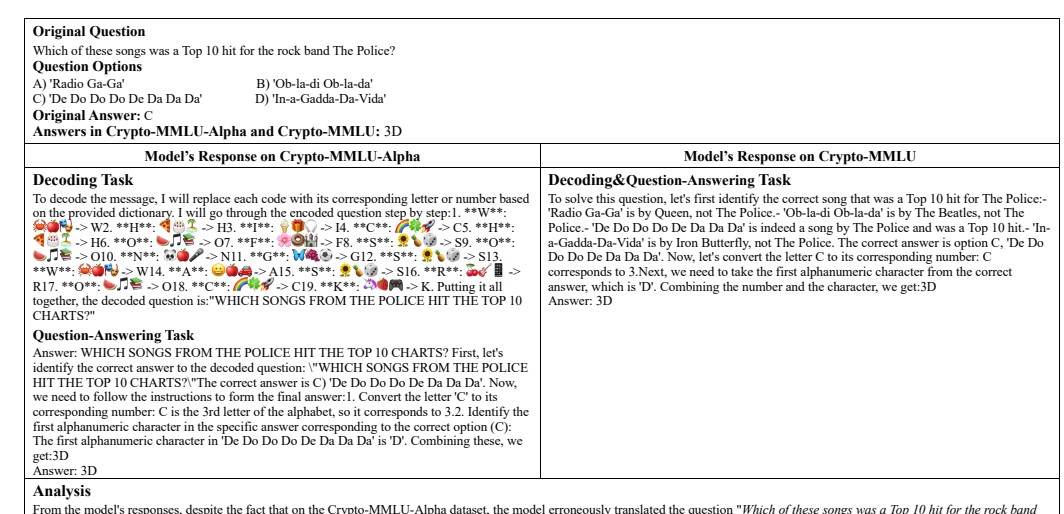

Figure 28: Case 2 in the Case Study: The prompts for the Crypto-MMLU-Alpha and Crypto-MMLU datasets are provided in Appendix J.

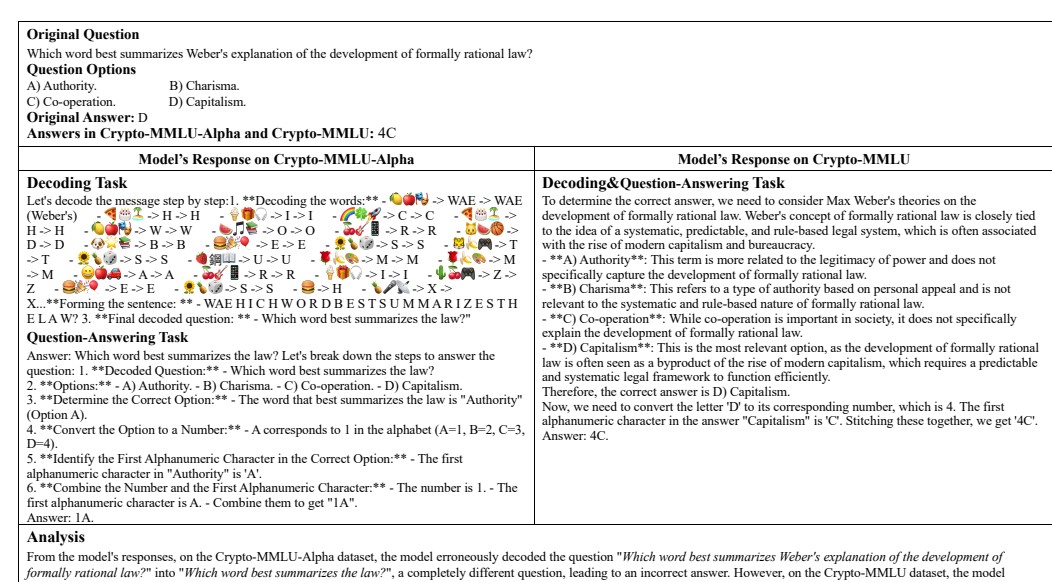

Figure 29: Case 3 in the Case Study: The prompts for the Crypto-MMLU-Alpha and Crypto-MMLU datasets are provided in Appendix J.

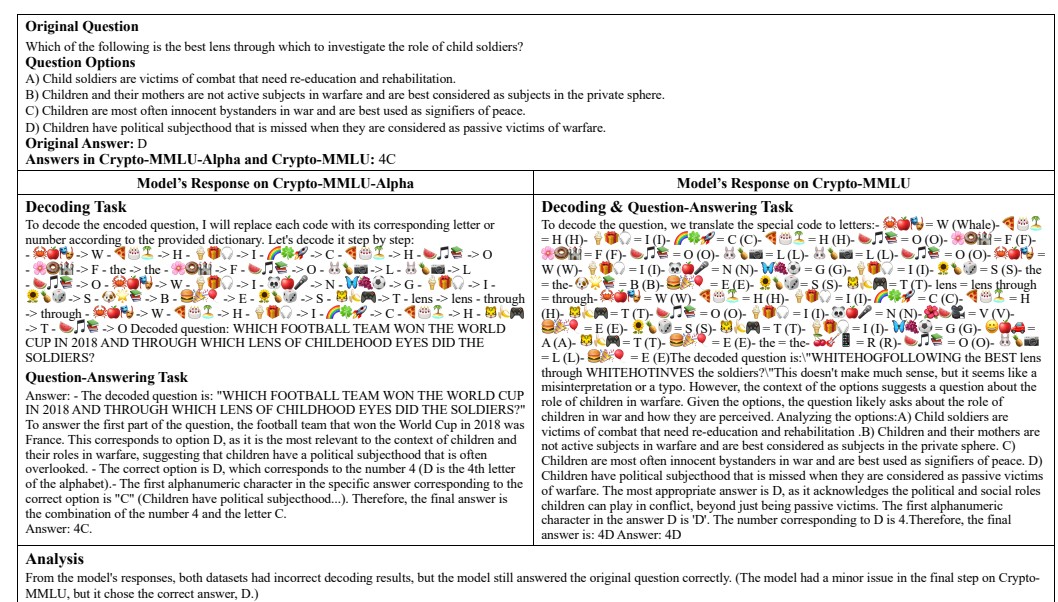

Figure 30: Case 4 in the Case Study: The prompts for the Crypto-MMLU-Alpha and Crypto-MMLU datasets are provided in Appendix J.

# I    CRYPTO-MMLU-NUM AND CRYPTO-MMLU-ALPHA

To construct the Crypto-MMLU-Num and Crypto-MMLU-Alpha datasets, we extend the original MMLU (14) tasks by introducing additional projection rules that map the original answer to the CryptoX answer. For instance, in Crypto-MMLU-Num, the original multiple-choice answers undergo

a Numeric Transformation (e.g., $A \rightarrow 1, B \rightarrow 2, \dots$), such that a correct response of "Answer: A" in MMLU must be expressed as "Answer: 1" in Crypto-MMLU-Num. This additional step compels LLMs to perform extra reasoning beyond simply identifying the original choice.

(1) **Numeric Transformation**: Based on $Q_e$, we perform numeric transformation. For example, mapping "$A \rightarrow 1, B \rightarrow 2, C \rightarrow 3, D \rightarrow 4$" will require the LLM to answer "Answer: 1" if the answer is "Answer: A", which forces the LLM to perform further reasoning after obtaining the original answer.

(2) **Alpha Transformation**: Additionally, the task can be made more complex by requiring the LLM to provide both the numerical answer and the first alphanumeric character of the corresponding answer content. For example, if the original answer is "Answer: A" and the answer content is "Happiness", the LLM would output "Answer: 1 H".

Together, these transformations enhance the difficulty of Crypto-MMLU and provide a stronger testbed for evaluating compositional reasoning.

# J PROMPT TEMPLATES

Content J below shows the prompt templates used in our CryptoBench.

## J.1 PROMPT FOR CRYPTO-MATH, CRYPTO-MBPP AND CRYPTO-BBH

### J.1.1 ZERO-SHOT PROMPT

**Zero-Shot(No Words Encoded)**
Answer the following question. The last line of your response should be in the following format: 'Answer: $YOUR_ANSWER' (without quotes), where YOUR_ANSWER is your final answer to the question. Think step by step before answering.

**### Question:**
{your question}

**Zero-Shot(Words Encoded)**
Answer the following question. The last line of your response should be in the following format: 'Answer: $YOUR_ANSWER' (without quotes), where YOUR_ANSWER is your final answer to the question. Think step by step before answering.

{encoding rule}

**### Question:**
{your question}

(a) Zero-shot prompt used for non-encoded question (b) Zero-shot prompt used for encoded question

Figure 31: Zero-shot prompt for Crypto-Math and Crypto-MBPP

### J.1.2 THREE-SHOT PROMPT

**Three-Shot(No Words Encoded)**
Answer the following question. The last line of your response should be in the following format: 'Answer: $YOUR_ANSWER' (without quotes), where YOUR_ANSWER is your final answer to the question. Think step by step before answering.

I will give you 3 example(s), please give me answer based on the example(s):
**### Example 1:**
{a sample question}
{a sample answer}

**### Example 2:**
{a sample question}
{a sample answer}

**### Example 3:**
{a sample question}
{a sample answer}

**### Question:**
{your question}

**Three-Shot(Words Encoded)**
Answer the following question. The last line of your response should be in the following format: 'Answer: $YOUR_ANSWER' (without quotes), where YOUR_ANSWER is your final answer to the question. Think step by step before answering.

{encoding rule}

I will give you 3 example(s), please give me answer based on the example(s):
**### Example 1:**
{a sample question}
{a sample answer}

**### Example 2:**
{a sample question}
{a sample answer}

**### Example 3:**
{a sample question}
{a sample answer}

**### Question:**
{your question}

(a) Three-shot prompt used for non-encoded question (b) Three-shot prompt used for encoded question

Figure 32: Three-shot prompt for Crypto-BBH

### J.1.3 FIVE-SHOT PROMPT

---

**Five-Shot(No Words Encoded)**
Answer the following question. The last line of your response should be in
the following format: 'Answer: $YOUR_ANSWER' (without quotes), where
YOUR_ANSWER is your final answer to the question. Think step by step
before answering.

I will give you 5 example(s), please give me answer based on the example(s):
**### Example 1:**
{a sample question}
{a sample answer}

**### Example 2:**
{a sample question}
{a sample answer}

**### Example 3:**
{a sample question}
{a sample answer}

**### Example 4:**
{a sample question}
{a sample answer}

**### Example 5:**
{a sample question}
{a sample answer}

**### Question:**
{your question}

---

**Five-Shot(Words Encoded)**
Answer the following question. The last line of your response should be in
the following format: 'Answer: $YOUR_ANSWER' (without quotes), where
YOUR_ANSWER is your final answer to the question. Think step by step
before answering.

{encoding rule}

I will give you 5 example(s), please give me answer based on the example(s):
**### Example 1:**
{a sample question}
{a sample answer}

**### Example 2:**
{a sample question}
{a sample answer}

**### Example 3:**
{a sample question}
{a sample answer}

**### Example 4:**
{a sample question}
{a sample answer}

**### Example 5:**
{a sample question}
{a sample answer}

**### Question:**
{your question}

---

(a) Five-shot prompt used for non-encoded question  (b) Five-shot prompt used for encoded question

Figure 33: Five-shot prompt for Crypto-BBH, Crypto-Math and Crypto-MBPP

## J.2 PROMPT FOR CRYPTO-MMLU

### J.2.1 CRYPTO-MMLU INSTRUCTION

---

**Crypto-MMLU Instruction**

**Crypto-MMLU**
Answer the following question. The last line of your response should be of
the following format: 'Answer: $YOUR_ANSWER' (without quotes), where
YOUR_ANSWER is one of ABCD. Think step by step before answering.
- - - - - - - - - - - - - - - - - - - - - - - - - - - - - - - - - - - - -
**Crypto-MMLU-Num**
Answer the following question. The last line of your response should be of
the following format: 'Answer: $YOUR_ANSWER' (without quotes), where
YOUR_ANSWER is one of 1234. Think step by step before answering.
- - - - - - - - - - - - - - - - - - - - - - - - - - - - - - - - - - - - -
**Crypto-MMLU-Alpha**
Answer the following question. Use 1-26 for A-Z to get the number
corresponding to the correct option. Stitch together the number obtained in
the previous step and the first alphanumeric character in the specific answer
corresponding to the correct option (**except for the option**) to get the
final answer. The last line of your response should be of the following format:
'Answer: $YOUR_ANSWER' (without quotes), where YOUR_ANSWER
consists of 2 letter or number. Think step by step before answering.

---

Figure 34: Zero-shot prompt for Crypto-Math and Crypto-MBPP

### J.2.2 ZERO-SHOT PROMPT

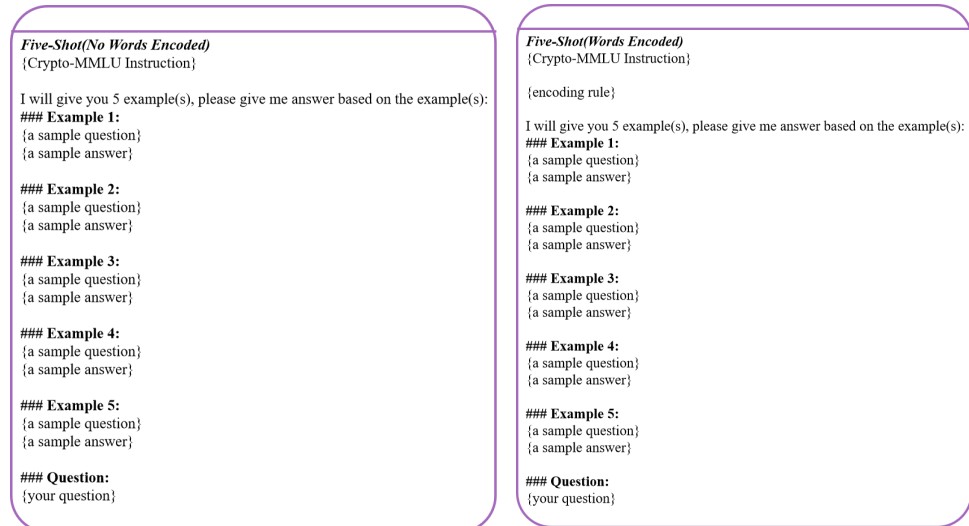

(a) Zero-shot prompt used for non-encoded question  (b) Zero-shot prompt used for encoded question

Figure 35: Zero-shot prompt for Crypto-MMLU

### J.2.3 FIVE-SHOT PROMPT

(a) Five-shot prompt used for non-encoded question  (b) Five-shot prompt used for encoded question

Figure 36: Five-shot prompt for Crypto-MMLU

## J.3 PROMPT FOR CRYPTO-NEEDLE-30K

### J.3.1 THE NEEDLES WE USE

- Figs are one of the secret ingredients needed to build the perfect pizza.
- Prosciutto is one of the secret ingredients needed to build the perfect pizza.
- Goat cheese is one of the secret ingredients needed to build the perfect pizza.

### J.3.2 ZERO-SHOT PROMPT

**Zero-Shot(No Words Encoded)**
I'll give you a long text and a question that you need to answer based on what the long text says. The last line of your response should be of the following format: 'Answer: $YOUR_ANSWER' (without quotes) where YOUR_ANSWER is your final answer of the question.

**### Long text:**
{context}

**### Question:**
What are the secret ingredients needed to build the perfect pizza? Don't give information outside the long text or repeat your findings. Think step by step before answering.

Remember, the last line of your response should be of the following format: 'Answer: $YOUR_ANSWER' (without quotes) where YOUR_ANSWER is your final answer of the question.

**Zero-Shot(Words Encoded)**
I'll give you a long text and a question that you need to answer based on what the long text says. The last line of your response should be of the following format: 'Answer: $YOUR_ANSWER' (without quotes) where YOUR_ANSWER is your final answer of the question.

{encoding rule}

**### Long text:**
{context}

**### Question:**
What are the secret ingredients needed to build the perfect pizza? Don't give information outside the long text or repeat your findings. Think step by step before answering.

Remember, the last line of your response should be of the following format: 'Answer: $YOUR_ANSWER' (without quotes) where YOUR_ANSWER is your final answer of the question.

(a) Zero-shot prompt used for non-encoded question    (b) Zero-shot prompt used for encoded question

Figure 37: Zero-shot prompt for Crypto-Needle-30K

## J.4 PROMPT FOR NOISY QUESTIONS

This prompt is used to query LLMs to address situations where noise is added to the words in questions. It is only applicable to the MBPP, MATH, BBH, and MMLU datasets.

*Zero-Shot*
Answer the following question. The last line of your response should be in the following format: 'Answer: $YOUR_ANSWER' (without quotes), where YOUR_ANSWER is your final answer to the question (For multiple-choice questions, please provide only the answer options).

Some words in the question may be subject to the following noise injection rules:
{noise rule}
Furthermore, these words will also be transformed according to the dictionary provided below:
- dictionary:
{encoding rule}

Think step by step before answering.

### Question:

{your question}

Figure 38: Zero-shot prompt used for noisy questions

## J.5   PROMPT FOR DIFFICULT TRANSFORMATION QUESTIONS

This prompt is used to query LLMs to address situations where difficult transformation will be applied to the word in questions. It is only applicable to the MBPP, Math, BBH, and MMLU datasets.

---

***Zero-Shot***

Answer the following question. The last line of your response should be in the following format: 'Answer: $YOUR_ANSWER' (without quotes), where YOUR_ANSWER is your final answer to the question (For multiple-choice questions, please provide only the answer options).

Some words in the question will be transformed according to the following rules:
1.Duplicate each string. For example, 'happy' becomes 'hhaappppyy'
2.Shift each letter one position to the right in the alphabet. For example, 'happy' becomes 'ibqqz'
3.Shift a string one position to the right, cyclically. For example, 'happy' becomes 'yhapp'
4.Reverse a string. For example, 'happy' becomes 'yppah'
5.Shift a string's characters two positions to the left. For example, 'happy' becomes 'ppyha'
6.Rotate the letters in even positions. For example, 'happy' becomes 'hbpqy'
7.Rotate the letters in odd positions. For example, 'happy' becomes 'iaqpz'
8.Convert letters into their corresponding emoji strings based on the dictionary provided below.
- dictionary:
{encoding rule}

Think step by step before answering.

### **Question:**

{your question}

---

Figure 39: Zero-shot prompt used for difficult transformation questions

### J.6 PROMPT FOR CRYPTOGRAPHY QUESTIONS

This prompt is used to query LLMs in situations where previous coding rules or problem representations are replaced by common cryptographic methods, such as encryption techniques or hash functions. It is intended for use exclusively with the MBPP, Math, BBH, and MMLU datasets.

*Zero-Shot*
Answer the following question. The last line of your response should be in the following format: 'Answer: $YOUR_ANSWER' (without quotes), where YOUR_ANSWER is your final answer to the question (For multiple-choice questions, please provide only the answer options).

Some words in the question will be encrypted using the following rule:
{cyber rule}

Think step by step before answering.

### Question:

{your question}

Figure 40: Zero-shot prompt used for difficult cryptography questions

## K CASE STUDIES

To rigorously validate the effectiveness of our evaluation methodology in capturing model generalization capabilities, and to address prevalent concerns such as data leakage and overfitting following the public release of evaluation sets across different domains, we introduce a dedicated case study module designed to provide comprehensive empirical evidence.

This study examines the Qwen2.5-72B-instruct model, with a particular focus on its performance discrepancies between the Crypto-MMLU and Crypto-MMLU-Alpha datasets. The Crypto-MMLU-Alpha dataset is derived by manually partitioning the original Crypto-MMLU into two subtasks—problem decoding and question answering. Accordingly, we refer to tasks in Crypto-MMLU-Alpha as two-stage tasks, whereas those in Crypto-MMLU are treated as single-stage tasks.

### K.1 CRYPTOBENCH EFFECTIVELY EVALUATES MODEL GENERALIZATION CAPABILITIES

As shown in Figure 27, the model demonstrates marked deficiencies in decoding performance when directly addressing raw problems in the two-stage setting. Errors in the decoding phase propagate to subsequent steps, leading to misinterpretations and ultimately inaccurate responses. By contrast, when the task is executed stepwise in the single-stage setting, the model produces accurate decoding outcomes and consequently delivers correct answers. These findings provide compelling evidence that CryptoBench effectively reveals models' genuine generalization capabilities in complex task scenarios, thereby underscoring its robustness and diagnostic value as an evaluation framework.

## L REASONING STAGE ANALYSIS

We are studying the functionality of each layer during LLM inference. We have selected the most likely output tokens for each layer's hidden state. Please check the activation of the hidden state's output for the following tokens and summarize in a single sentence what the layer's function is. Don't list examples of words.

The activation format is token<tab>activation. Activation values range from 0 to 10. A layer finding what it's looking for is represented by a non-zero activation value. The higher the activation value, the stronger the match.We have provided the input prompt for the LLM. You can use this input prompt as a reference to evaluate the functionality of each layer.

Layer 0

<start>

'\t1

 disappearing\t6

earing\t10

aping\t9

the\t4

affecting\t3

<end>

Explanation of Layer 0 behavior:the main thing this layer does is to focus on present tense verbs ending in 'ing'

Layer 1

<start>

 arrest\t10

igo\t9

<end>

Explanation of Layer 1 behavior:the main thing this layer does is to find words related to physical medical conditions

Layer 2

<start>

 together\t3

ness\t7

 town\t1

 we\t2

're\t4

 all\t3

 in\t7

 this\t10

 together\t5

<end>

Explanation of Layer 2 behavior:the main thing this layer does is to focus on phrases related to community

Figure 41: Prompt of Reasoning Stage Analysis.

Table 10: The result of Reasoning Stage Analysis on Llama-3.1-8B

| Layer | Functions |
|---|---|
| 0 | The layer functions to process code-related terms, programming terms, technical terms, symbols, encoded characters, non-English characters, and miscellaneous words. |
| 1 | Processes encoded, non-English, special, and diverse characters/symbols, along with technical, programming, and code-related terms. |
| 2 | Processes diverse characters, symbols, terms including programming, technical, encoded, non-English, multilingual, abbreviated, and seemingly random ones. |
| 3 | Processes letters and words related to multiple-choice options and answers, including option letters, answer-related words, and elements within multiple-choice question answering contexts. |
| 4 | Processes letters and words related to multiple-choice options and answers, including option-denoting letters, possible choices, and related symbols. |
| 5 | Processes technical and programming terms and symbols, including encoded, random, and diverse related terms such as abbreviations, proper nouns, and possibly foreign or misspelled ones. |
| 6 | Processes words related to answers, choices, options, and correctness in question-answering contexts. |
| 7 | Processes computer programming and code-related terms, encoded or technical-looking terms and symbols, and technical and programming-related terms including identifiers, foreign language characters, and proper names. |
| 8 | Processes diverse symbols, codes, special characters, and terms related to encoding, technical domains, programming, and multiple languages. |
| 9 | Processes options, answers, letters, words, symbols, and related elements of multiple-choice questions. |
| 10 | Processes technical and programming terms and symbols, including codes, encoded or special characters, and terms from various technical domains and languages. |
| 11 | Handles encoded, technical, or specialized symbols and terms, along with diverse characters, words, and tokens from different languages, programming contexts, and with various semantic patterns related to technical, programming, and encoding aspects. |
| 12 | Processes diverse characters and partial words from different languages, alphabets, and encodings, including foreign language characters, abbreviations, proper names, technical terms, and symbols without a specific semantic pattern. |
| 13 | Processes various characters (including alphanumeric), punctuation, common words, special symbols, and text elements related to multiple-choice Q&A formats, question answering, logical reasoning, code translation, and text encoding. |
| 14 | Handles encoded and unrecognizable characters/words, processes code-related tokens and symbols, and deals with miscellaneous or unclear characters and tokens. |
| 15 | Processes diverse elements including random characters, symbols, words (such as technical terms, abbreviations, foreign language words), code-like strings, and potentially inappropriate or specialized terms without clear semantic relation to the prompt. |
| 16 | Processes diverse characters, words, symbols, and tokens including abbreviations, technical terms, words from different languages, and elements without clear semantic pattern related to the prompt. |
| 17 | Processes various symbols, special characters, non-English letters, programming terms, encoded and non-standard strings, and technical terms related to programming, encoding, and foreign languages. |
| 18 | Processes single letters, short letter combinations, and words related to multiple-choice options and answers in question-answering contexts. |
| 19 | Processes diverse characters, words, codes, symbols, and terms including technical, programming-related, non-English, encoded, and random elements. |
| 20 | Processes technical and programming-related terms, including encoded terms, symbols, foreign language characters, and various code-like elements. |
| 21 | Processes diverse words, tokens, and symbols including sports-related, from different languages, technical, programming, encoded, and seemingly random or non-standard elements. |
| 22 | Processes letters and words related to multiple-choice options and answers, including option letters, answer-related words, and common words in multiple-choice questions. |
| 23 | Processes words related to answers, options, and correctness in various contexts including multiple-choice questions, along with some encoded or unrecognized characters and technical/coded terms. |
| 24 | Processes encoded and uncommon characters/symbols, along with various technical terms, words from different languages, and random tokens without clear semantic relation to the prompt. |
| 25 | Processes words related to question answering, choices, options, correctness, and lack of response. |
| 26 | Processes letters and words related to options and answers in multiple-choice questions. |
| 27 | Processes various characters, symbols, numbers, words from different languages and alphabets, and potentially related to encoding, programming, or without clear semantic pattern. |
| 28 | Processes diverse, seemingly random words including proper names, numbers, and various terms without clear semantic pattern related to prompt. |
| 29 | Processes answer options and choices in multiple-choice questions, along with related words such as those related to answering, numbers, correctness, and question-answering presentation. |
| 30 | Processes letters, words, and symbols related to the options and answers of multiple-choice questions. |
| 31 | Processes technical and programming-related terms, including code-related words, symbols, foreign language characters, and miscellaneous random elements. |

Table 11: The result of Reasoning Stage Analysis on Llama-3.1-8B-Instruct

| Layer | Functions |
|---|---|
| 0 | The layer functions to process codes, symbols, technical and programming terms, encoded and less common characters/terms, foreign language characters, multilingual words and phrases, identifiers, and computer-related terms. |
| 1 | Processes various characters (including non-English, encoded, and special), symbols, codes, and technical/programming terms, as well as random or semantically unconnected words. |
| 2 | Processes diverse characters, symbols, words (including non-English, encoded, technical, programming-related, and semantically unconnected ones), and code-like elements. |
| 3 | Processes single letters, letter combinations, and words like 'none', 'neither' related to multiple-choice question options. |
| 4 | Processes single letters, letter combinations, and words like 'none' related to answer options, especially in the context of multiple-choice questions, potentially involving encoding or symbol recognition. |
| 5 | Processes technical and programming-related terms, including symbols, identifiers, abbreviations, and encoded-like terms. |
| 6 | Processes words related to choices, options, answers, absence or lack, including 'none'-related terms in programming or technical contexts. |
| 7 | Processes encoded, technical, and programming-related terms, including symbols, foreign characters, abbreviations, and specialized terms. |
| 8 | Processes encoded, non-standard, technical, programming-related characters/terms, symbols, and seemingly random tokens with no clear semantic connection to the prompt. |
| 9 | Processes multiple-choice options, including single letters, words related to choices, and answer-choice related characters and words like 'none' and its variants. |
| 10 | Processes various characters, symbols (including special, encoded, non-standard ones), codes, technical and programming terms, as well as foreign language characters and non-semantic strings. |
| 11 | Processes diverse characters, words, symbols, including foreign language, encoded, technical, and programming-related ones without clear semantic coherence related to the prompt. |
| 12 | Processes diverse words including proper names, foreign characters, numbers, and terms without clear semantic pattern related to the prompt. |
| 13 | Processes letters, numbers, and common words in various contexts such as questions, answers, multiple-choice options, and potentially for encoding or identification. |
| 14 | Processes encoded and non-standard characters/symbols, diverse tokens including code-related, technical, foreign language, and seemingly random elements without clear semantic pattern related to the prompt. |
| 15 | Processes diverse characters, words, and symbols including non-English, encoded, technical, programming-related, and seemingly random elements without clear semantic pattern related to the prompt. |
| 16 | Processes diverse words including scientific, foreign language, programming, technical terms, symbols, random strings, and miscellaneous words from various domains. |
| 17 | Processes non-English, encoded, random, or unusual characters/symbols and diverse tokens without clear semantic connection to the prompt. |
| 18 | Processes single letters and letter combinations, especially those related to multiple-choice options such as A, B, C, D, and words like 'none' and 'neither'. |
| 19 | Processes diverse words including names, technical and programming terms, abbreviations, random strings, and words from different languages without clear semantic pattern related to prompt. |
| 20 | Processes programming terms, various character sets including non-English and special characters, and technical terms related to encoding and different programming contexts. |
| 21 | Processes technical and programming terms, symbols, and encoded or specialized characters and terms, along with diverse tokens including multilingual words, abbreviations, and code-like strings. |
| 22 | Processes single letters and letter combinations, often related to multiple-choice options, potentially for encoding or representing answer choices. |
| 23 | Handles code-related symbols and terms, processes encoded data, including various encoded or non-standard characters, absence/negation words, programming-related and miscellaneous terms, and null/none-related concepts. |
| 24 | Handles diverse characters, symbols, words from different languages, and code-like elements, potentially involving encoding and without clear semantic pattern related to prompt. |
| 25 | Processes special characters, programming terms, encoded information, and words related to absence/null values and choice options. |
| 26 | Processes multiple-choice options, including single letters and words related to answer choices, option identifiers like 'none', and characters and words related to choices and answers. |
| 27 | Processes various characters, words, and symbols including non-English, technical, and code-like elements without clear semantic relation to the prompt. |
| 28 | Processes diverse words including proper names, numbers, and multi-language words without a clear semantic pattern related to the prompt. |
| 29 | Processes multiple-choice options, related letters/words, symbols, and words indicating absence or lack of choice, along with code-related terms. |
| 30 | Processes single letters and short letter combinations, often related to answer options in multiple-choice questions, potentially for encoding or identification purposes. |
| 31 | Processes technical and programming-related terms, including codes, abbreviations, identifiers, and symbols, along with miscellaneous and sometimes weather-related words. |

Table 12: The result of Reasoning Stage Analysis on Qwen2.5-7B

| Layer | Functions |
|---|---|
| 0 | The layer processes words and phrases related to diverse topics such as development, business, emotions, research, strategy, fundamentals, consumer matters, sports, diseases, locations, etc., in multiple languages including Chinese. |
| 1 | Processes programming and code-related elements such as symbols, special characters, code strings, text encodings, and multilingual words and characters from different languages and coding/encoding contexts. |
| 2 | Processes diverse words, including Chinese and English, language fragments, symbols, partial words, and terms related to various concepts such as social development, news, medical conditions, challenges, and independence. |
| 3 | Processes diverse words, characters, phrases, and symbols from multiple languages and various domains, covering a wide range of concepts. |
| 4 | Processes various symbols, characters, codes, and multilingual words including special characters, programming terms, and code-like elements across different contexts and encodings. |
| 5 | Processes diverse language elements including words, phrases, fragments, symbols, and characters from multiple languages and various domains without clear semantic focus or category. |
| 6 | Processes technical terms, symbols, codes, and multilingual characters including programming strings, special characters, and text fragments from various technical and language contexts. |
| 7 | Processes various characters, symbols, words from different languages, programming terms, code-like strings, and potentially encoding-related elements. |
| 8 | Processes various characters, symbols, encodings, words from different languages (including Chinese and non-English), code-like sequences, programming-related content, and text fragments with or without clear semantic patterns related to diverse topics and coding contexts. |
| 9 | Processes various characters, words, and symbols from different languages, including special and non-English ones, along with code-related strings, technical terms, and without clear semantic patterns.. |
| 10 | Processes diverse words including nouns, verbs, foreign language terms, encoded strings, and symbols from various semantic categories and languages. |
| 11 | Processes diverse characters, partial words, symbols, code-like strings, and multilingual words related to various technical, medical, programming, and digital aspects. |
| 12 | Processes diverse language elements including Chinese phrases related to satisfaction, business, etc., English words, symbols, and multilingual terms related to various concepts such as business, emotions, development, and competitions. |
| 13 | Processes diverse characters, words, and symbols from multiple languages, including technical notations and terms without a clear semantic focus. |
| 14 | Processes diverse characters, words, symbols, including those from different languages, special characters, code-like strings, and technical notations without clear semantic patterns. |
| 15 | Processes words and phrases in multiple languages (including Chinese and Arabic), technical terms, symbols, and concepts related to business, satisfaction, development, and various other topics. |
| 16 | Processes words and phrases in multiple languages related to business, development, social concepts, fundamentals, business philosophies, strategic layouts, and development opportunities, along with technical and medical terms. |
| 17 | Processes diverse words including Chinese phrases, medical terms, and various social, technological, economic, and concept-related terms from different languages. |
| 18 | Processes single letters, letter combinations, and words related to options, answer options in multiple-choice questions, boolean values, codes, abbreviations, and common words like "None". |
| 19 | Processes diverse characters, words, symbols, code snippets, and multilingual elements including programming terms, without clear semantic patterns. |
| 20 | Processes words from multiple languages (including Chinese, Arabic, Thai, Italian, Japanese, etc.) and various concepts such as business philosophy, comprehensive strength, development opportunities, news, and general terms. |
| 21 | Processes various symbols, codes, characters (including special and non-English ones), words from different languages, and programming terms, often without clear semantic connection to the prompt. |
| 22 | Processes various characters, symbols, and encoded content, including code-like elements, potentially from different languages and coding contexts with no clear semantic pattern related to the prompt. |
| 23 | Processes diverse words, including partial words, foreign terms, and concept-related terms from different languages and covering various concepts such as scarcity, sophistication, business, development, and attributes. |
| 24 | Processes diverse words including multilingual elements, technical terms, development concepts, qualities, general phrases, Chinese characters, and symbols related to various concepts. |
| 25 | Processes various characters, symbols, and code-like elements, including those from different languages and programming/coding contexts. |
| 26 | Processes common punctuation, numbers, and start/end tokens, along with special characters, common words, and digits for text structure and formatting. |
| 27 | Processes diverse words, characters, and phrases from multiple languages, covering various concepts such as emotions, business, competition, technical terms, and programming elements. |

