# OpenReview forum: "CryptoX : Compositional Reasoning Evaluation of Large Language Models"
_ICLR.cc/2026/Conference — ICLR 2026 Conference Withdrawn Submission_

### Official Review · Reviewer_pgz1 · 2025-10-27

**Soundness:** 3
**Presentation:** 2
**Contribution:** 2
**Rating:** 4
**Confidence:** 3

**Summary:**

This paper presents CryptoX, a framework to evaluate large language models’ compositional reasoning by embedding cipher-like transformations into tasks. Through the benchmark CryptoBench, the authors test 40+ models and show that performance depends mainly on task decomposition and subtask solving, while final integration is less critical. They further analyze model activations to reveal a stage-wise process—rule understanding, subproblem solving, and answer synthesis—arguing that CryptoX offers a systematic way to probe this distinct reasoning ability.

**Strengths:**

1.The paper introduces a novel and systematic evaluation framework **(CryptoX)** for compositional reasoning, with a carefully designed benchmark (CryptoBench) that allows controlled difficulty scaling and clear attribution of model failures. This provides a reproducible and extensible way to study an important but underexplored capability of LLMs.

2.The authors conduct comprehensive experiments across 40+ open- and closed-source models, combined with mechanistic interpretability analysis (logit lens, neuron activation) to reveal stage-wise reasoning dynamics. This breadth and depth of analysis strengthen the validity of their findings and provide useful insights for the community.

**Weaknesses:**

1.**Reasonableness of Task**: DesignCryptoX's rules are artificially constructed (encryption/transformation). Can they represent real-world "combinatorial reasoning"? I'm concerned that the task is too symbolic and may lack generalizability.

2.**Writing Issues**: The paper suffers from clarity problems in terminology and exposition. Key definitions (e.g., Crypto-MMLU, -Num, -Alpha) are scattered across the appendix rather than consolidated in the main text, which forces readers to cross-reference. Figures such as Figure 4 are insufficiently explained, with captions that lack self-contained definitions. Some conceptual claims (e.g., “A+B vs A×B”) are not clearly tied to concrete examples or experimental evidence, making the argument harder to follow. Overall, the writing would benefit from tighter organization, clearer figure captions, and centralizing critical definitions in the main body rather than in supplementary sections.

**Questions:**

My problem mainly lies in the limited representativeness of the CryptoBench task design. CryptoBench relies mainly on synthetic encryption/translation rules with explicit prompts. This design risks oversimplifying compositional reasoning into “applying known rules,” which may not transfer well to naturalistic reasoning scenarios involving noise, partial rules, or cross-domain knowledge.

---

### Official Review · Reviewer_w6U9 · 2025-10-29

**Soundness:** 2
**Presentation:** 2
**Contribution:** 2
**Rating:** 4
**Confidence:** 3

**Summary:**

The paper introduces a compositional reasoning benchmark for LLMs that combines decryption with downstream task solving. In this setup, instructions from existing benchmarks are encrypted using rule-based methods, requiring models to first decrypt the prompt and then perform the original task. The encryption rules are provided alongside the problem, making each instance self-contained apart from the base benchmark. Two types of encryption rules are used: codebook-based text replacement and programmatic text modification (e.g. repetition of characters). Performance is summarized using Area-Upon-Curve (AUC), which measures model accuracy as encryption complexity increases. Experimental results indicate that the encrypted, composite tasks are consistently more difficult than the original ones across various LLMs. Additional analysis suggests that models implicitly separate the process into decryption and question answering, reflecting the intended compositional structure.

**Strengths:**

-	Originality & Significance: The approach of extending existing benchmarks to compositional reasoning is interesting, as it aims to balance controllable reasoning complexity with practical downstream tasks. When properly executed, this framework could provide an effective framework for analyzing LLM compositional reasoning behavior.
-	Quality: The experiments include a wide range of evaluated models, which strengthens the credibility of the results.
-	Clarity: The results in Section 4.1(2) demonstrate a nontrivial performance gap between isolated and composite task runs, supporting the benchmark’s validity as an evaluation tool for compositional reasoning.

**Weaknesses:**

-	The definition of compositionality in this study is too limited. Each example follows a fixed two-step process of decryption followed by question answering, with no exchange of information between them. Because the reasoning path is identical across all samples and the reasoning graph connects only a single pair of tasks (decryption and downstream reasoning), the task becomes a static routine rather than a test of adaptive decomposition. The task type itself is also limited since the encryption is implemented only as text replacement. As a result, the setup does not effectively evaluate compositional reasoning in LLMs, where the main challenge should be dynamically identifying and integrating subtasks.
-	I doubt that AUC is an appropriate metric for evaluating compositionality. First, the number of encoded words k has not been validated as an indicator of compositional complexity. Second, because the benchmark does not test multiple encryption levels for the same downstream instance, the resulting AUC curve across k cannot be meaningfully interpreted: each point represents a different task rather than a controlled change in compositionality. Third, the metric fails to capture qualitative differences in the compositionality curve. For instance, a model that performs well for k ≤ N but drops to zero beyond that and another that declines steadily across all k values could have similar AUCs, even though their compositional reasoning behaviors are fundamentally different. The metric offers no way to distinguish between them.
-	Figure (a) is hard to grasp. If the point is to compare indivisual eval (A or B) with composed task (A + B), there would be more straightforward way of depicting this.
-	Figure (b) might be misleading, since it is expected that composing the downstream task (MMLU) with another task (decryption) would result in performance reduction and it could be too hasty a conclusion to interpret this as low compositional reasoning capacity.
-	Minor point: “We then conduct three complementary analyses” is repeated in line 80.
-	Minor point: whitespace omitted after (2) and (3) in lines 84 and 86, respectively.
-	Minor point: typo in Figure 2 plot: Ruls -> Rules

**Questions:**

-	From the experiment setup, only BBH is evaluated with few-shot prompting. I presume that it would be to follow standard practice in the original paper, but it would be better to state this up front to prevent potential confusion.

---

### Official Review · Reviewer_Y2Mv · 2025-10-30

**Soundness:** 2
**Presentation:** 3
**Contribution:** 2
**Rating:** 4
**Confidence:** 4

**Summary:**

The CryptoX paper presents an interesting method to explore the reasoning ability of models via the lens of Compositional Reasoning. The authors present a way of modifying existing benchmarks into a format that requires multiple stages of unpicking to recover the original question, and then evaluates a wide range of models. This approach has two nice features. First, the baseline is easily understood as the benchmarks used as familiar in the literature. Second, the degree of encryption is controllable and therefore can be tuned + applied programmatically. Generally the work is strong, and my main points of concern that I’d like to see addressed revolve around further evidence to show that solving the multistage problems demonstrates a genuine compositional reasoning and is different to robustness against noisy / corrupt inputs.

**Strengths:**

- The authors present a scalable method to take existing benchmarks for LLMs and encrypt them with a combination of transformations that happen at a word level, and then whole scale replacement with moise / emoji code. This makes it easy to apply to any existing benchmark

- 4.1 Analysis (2) attempts to disentangle the one step vs combined reasoning. This is exactly the sort of analysis I was hoping to see, and is a very good start to addressing the concern I raise in the questions.

- Analysis 4.3 seems to be the most important evidence that this actually tests the multiple stages of reasoning, and again is really good to see.

- The extent of coverage of models is impressive, as is the coverage of supplementary results in in the appendix.

**Weaknesses:**

- The key weakness I find in the paper is a lack of evidence that the proposed method tests for compositional reasoning rather than robustness to noise. I appreciate the authors attempts in this direction that are already provided, but I would like to see some further evidence to address specific concerns. Showing more evidence that this method is meaningfully different to noise robustness would go a long way to convincing me that this proposed method is novel and useful.


“As shown in Table 2, increasing the number of encoded words consistently raises reasoning difficulty and lowers evaluation metrics for all models. The fact that this pattern holds across diverse domains and benchmarks suggests a systematic issue rather than a domain specific one, thereby demonstrating that CryptoX reliably reflects models’ true CR ability even on the easiest benchmarks.” ([“CryptoX : Compositional Reasoning Evaluation of Large Language Models”, 2025, p. 5](zotero://select/library/items/BM6TILRF)) ([pdf](zotero://open-pdf/library/items/22PE2T8X?page=5))

- I use this quote as an example - I would expect that 2x or 3x ing the length of the sequence with a lot of random characters would degrade the model performance. Without a baseline of taking the same number of words from the question and replacing them with random characters it is really hard to support this claim that the compositional nature of the encoding is meaningfully different to adding random noise.

- Additionally when I look at the example prompts the obvious question is can you basically just read what the question is by ignoring the emojis? In a lot of cases you could make a good guess, and I suspect the model could too. My point is that guessing the correct missing part of the question is different to unpicking the question. I wonder if the authors could try a few analyses:

    - Replace the encrypted words with random tokens ~ the same length as the original vs the length of the emoji version

    - Replace random tokens throughout the question, not just specific words - this would make it harder to see the original form of the question, and require more initial task to recover?

    - (More extensive - if you see utility) Provide a key to the model so it can translate, and then encrypt the whole question. This would require a full task1 -> task2, and would leak less information from the initial state I suspect?

**Questions:**

- Do you have a plot to compare the impact of transforming X% of the sequence using a single method and X% of the sequence via a multi step set of transformations? I’m curious about the % of noise introduced, vs the complexity of the noise? I’d like to see a plot of score vs X% corruption of sequence via single and multiple steps?

- With the noisy rule and transformation rule - do you have a quantification of how much this degrades the output? I would expect the base models to be robust to spelling mistakes (noisy) and to a moderate degree the transformations. Would it be possible to add a baseline where the same number of words are removed and just replaced with random tokens?

- I’m interested to see the impact of the emoji insertions? I would imagine in this context the model largely ignores them? As a person you can still largely read the question in the example? Can you show some example attention patterns to see if the model is just ignoring the emojis?

- Can the authors add results for GPT 5 and Claude 4.1? The omission of these feels notable in Table 2, Fig 3 and Fig 4. I understand these are relatively recent models but it would strengthen the evidence to include these latest models if possible before publication.

- In analysis 4.2 how are you mapping the stages onto the layers of the model? Why does Llama reach stage 2 after 20 layers while Qwen reaches it after 6? This feels like a really substantial claim to be making about what each of the portions of the model are doing but I don’t follow how you’ve defined this? Please can you expand on how you have done this? How do you know these layers in the model are doing this? If you can back this up this is excellent, I don't feel quite convinced yet, but I'd love some more evidence of this!


Nitpicks:

- Echoing reviewer w6U9 the reference standard is not correct - please correct this and update the manuscript - and be careful about the page limit - additionally I’d recommend sorting the references so they appear in the bibliography in the order they first appear in the text.

- “We then conduct three complementary analyses and obtain the following findings: We then conduct three complementary analyses:” ([“CryptoX : Compositional Reasoning Evaluation of Large Language Models”, 2025, p. 2](zotero://select/library/items/BM6TILRF)) ([pdf](zotero://open-pdf/library/items/22PE2T8X?page=2))

    - Duplicate phrase

- Please remake figure 6 at an appropriate figure size for the page!

---

### Official Review · Reviewer_KDNH · 2025-10-31

**Soundness:** 2
**Presentation:** 2
**Contribution:** 1
**Rating:** 2
**Confidence:** 4

**Summary:**

The paper introduces CryptoX, a plug-in evaluation framework that evaluates compositional reasoning (CR) ability of LLMs. In detail, CryptoX modified the question by encrypting it. Building on CryptoX, they develop CryptoBench, a suite of benchmarks with encrypted instructions. They introduce AUC as evaluation metric and evaluate over 40 popular LLMs. After that, they perform extensive analysis.

The contributions are:

1. The authors introduce CryptoX, an evaluation method for CR of LLMs.
2. The authors construct CryptoBench, and evaluate over 40 LLMs.
3. The authors claim CR process can be divided to three stages, and CR ability can be characterized in two dimensions.

**Strengths:**

The method and idea are straightforward and easy to understand.

The authors conduct extensive experiments.

Experiment details, including prompts and codes, are well demonstrated and shared.

**Weaknesses:**

1. This paper narrows down the topic to deciphering, making the research question less important. (not representative and far from the real world) A reasonable benchmark should be representative and evaluate a general ability.

2. Some claims are made ambiguously. For example, "the true CR ability", how to define true. Even with the explanation in Section 4.1, readers may still be confused about "true".

3. The analysis shares little value to the broader ICLR community. The results in Sections 4.1, 4.2, and 4.3 are obvious.

**Questions:**

See weakness.

---

### Note · Authors · 2026-01-22

I have read and agree with the venue's withdrawal policy on behalf of myself and my co-authors.